



# Deep ice as a geochemical reactor: insights from iron speciation and mineralogy of dust in the Talos Dome ice core (East Antarctica)

Giovanni Baccolo[1,2], Barbara Delmonte[1], Elena Di Stefano[1,2,3], Giannantonio Cibin[4], Ilaria Crotti[5,6], Massimo Frezzotti[7], Dariush Hampai[8], Yoshinori Iizuka[9], Augusto Marcelli[8,10], and Valter Maggi[1,2]

[1]Environmental and Earth Science Department, University Milano-Bicocca, Italy
[2]Istituto Nazionale di Fisica Nucleare, section of Milano-Bicocca, Milan, Italy
[3]Department of Physical, Earth and Environmental Sciences, University of Siena, Italy
[4]Diamond Light Source, Harwell Science and Innovation Campus, Didcot, UK
[5]Department of Environmental Sciences, Informatics and Statistics, Ca' Foscari University of Venice, Italy
[6]Laboratoire des Sciences du Climat et de l'Environnement IPSL/CEA-CNRS-UVSQ UMR, Gif-sur-Yvette, France
[7]Department of Science, University Roma Tre, Italy
[8]Istituto Nazionale di Fisica Nucleare, Laboratori Nazionali di Frascati, Frascati, Italy
[9]Institute of Low Temperature Science, Hokkaido University, Sapporo, Japan
[10]Rome International Center for Materials Science - Superstripes, Rome, Italy

**Correspondence:** Giovanni Baccolo (giovanni.baccolo@unimib.it)

**Abstract.** Thanks to its insolubility, mineral dust is considered a stable proxy in polar ice cores. With this study we show that below an ice-depth of 1000 m, the Talos Dome ice core (Ross Sea sector of East Antarctica) presents evident and progressive signs of post-depositional processes affecting the mineral dust records. We applied a suite of established and cutting edge techniques to investigate the properties of dust present in the Talos Dome ice core, ranging from concentration and grain-size to
5  elemental-composition and Fe-mineralogy. Results show that through acidic/oxidative weathering, the conditions of deep ice at Talos Dome promote the dissolution of specific minerals and the englacial formation of others, deeply affecting dust primitive features. The expulsion of acidic atmospheric species from ice-grains and their concentration in localized environments is likely the main process responsible for englacial reactions and is related with ice re-crystallization. Deep ice can be seen as a "geochemical reactor" capable of fostering complex reactions which involve both soluble and insoluble impurities. Fe-bearing
10  minerals can efficiently be used to explore such transformations.

## 1 Introduction

Antarctic ice is a valuable archive which allowed to reconstruct the climatic history of Earth during the last 800.000 years through ice cores (Wolff et al., 2010) and even further back in time thanks to ancient ice outcropping at blue-ice fields (Yan et al., 2019). Mineral dust is one of the many proxies that is possible to investigate in ice cores. Its importance stems from
15  its role in Earth's climate system: production, transport and deposition of dust are controlled by climate-related processes, but at the same time dust affects the climate (Maher et al., 2010). Ice cores have been fundamental for understanding the role of atmospheric dust in past climatic periods. Studying the properties of dust trapped in ice cores, it has been possible to retrieve





information about the influence of climate on the dust cycle (Delmonte et al., 2004) and about the effects of dust under different climatic regimes (Mahowald et al., 1999; Wolff et al., 2006; Potenza et al., 2016).

It is known that the physical and compositional properties of dust trapped in ice cores are influenced by climatic, environmental and atmospheric processes (Sugden et al., 2009; Delmonte et al., 2017; Markle et al., 2018). The concentration of dust in ice is strictly controlled by climate (Delmonte et al., 2002), while grain size is related to its atmospheric transport (Delmonte et al., 2017; Albani et al., 2012a) and geochemistry to dust sources (Delmonte et al., 2004).

In the last years a growing number of studies focused on the fact that the proxies used to infer paleoclimatic information from ice cores are not imperishable. They are affected by alterations occurring after their incorporation into snow which are defined as post-depositional processes. As a general rule, the older the ice and the stronger the influence of such changes, despite some occur in the surficial part of ice sheets, such as loss of volatile species, fractionation in firn and wind-scouring (Blunier et al., 2005; Town et al., 2008; Buizert et al., 2013).

Post-depositional processes recognized to act in deep ice are related to three main causes: 1- the interaction between ice flow and bedrock (Goossens et al., 2016); 2- the metamorphism of ice (Faria et al., 2010); 3- the diffusion of impurities (Barnes et al., 2003). An irregular bedrock can produce stratigraphic disturbances affecting the original ice succession for several hundred meters. On the contrary, ice metamorphism and englacial diffusion act at a smaller scale, comparable to the size of ice-crystals (mm-cm scale) (Rempel et al., 2002; Faria et al., 2010). The effects of such phenomena on the proxies investigated in ice cores are not yet fully understood, but improving their comprehension will be topical considering the "Oldest Ice" challenge (Fischer et al., 2013). One aspect that still needs to be addressed is the role of depth and pressure in post-depositional processes. From the top to the bottom of Antarctic ice cores, temperature-pressure conditions change significantly, promoting different and diverse physical and/or chemical reactions (Tison et al., 2015).

Dust is considered relatively immobile and stable in ine and its concentration has been used to synchronize deep ice cores when other proxies were deteriorated (Ruth et al., 2007; Kawamura et al., 2017). But mineral particles are not untouched by post-depositional changes. Impurities in ice are affected by the small-scale relocation resulting from ice metamorphism, which in deep ice consists in the growth and re-crystallization of ice grains (Faria et al., 2010). Being most of the impurities incompatible with respect to the ice lattice (Wolff, 1996), during the re-crystallization they are expulsed from the ice crystalline structure and accumulated at ice grain junctions or within intra-grain micro-inclusions (Mulvaney et al., 1988; De Angelis et al., 2005; Sakurai et al., 2017; Stoll et al., 2021). The accumulation of soluble and insoluble impurities forms eutectic mixtures whose pressure melting point is below the ice temperature, thus allowing for the local formation of liquid brines which favor *in situ* chemical reactions (Fukazawa et al., 1998; De Angelis et al., 2005, 2013; Sakurai et al., 2017). Such small scale environments are dominated by sulphur-rich acidic species, strongly affected by re-mobilization and concentration because of their high incompatibility with the ice lattice (Mulvaney et al., 1988; Fukazawa et al., 1998; Wolff, 1996). The interaction between acidic brines and concentrated impurities, including dust, leads to acid-base reactions (Traversi et al., 2009; Ohno et al., 2016). With respect to dust the most common reactions which have been recognized to take place into ice are the dissolution of carbonates, the precipitation of gypsum (Ohno et al., 2006; Iizuka et al., 2008; Eichler et al., 2019) and of other uncommon sulphates (Ohno et al., 2014), or the englacial formation of secondary iron minerals (De Angelis et al., 2013; Baccolo et al.,





2021). In very deep englacial environments also the re-precipitation of carbonates have been reported, suggesting that once acidity is consumed, additional reactions take place (Tison et al., 2015).

The iron fraction of dust is particularly sensitive to englacial transformations. De Angelis et al. (2013) identified secondary Fe-bearing minerals in the deep part of the EPICA Dome C ice core, while Eichler et al. (2019) identified in the EPICA Dronning Maud Land ice core a few particles whose Raman signature was compatible with jarosite, a Fe-K sulphate forming through acidic weathering (Papike et al., 2006). This finding has been confirmed by a thorough investigation of TALDICE, where jarosite has been found below the depth of 1000 m and interpreted as the result of acidic, water-limited weathering of

dust (Baccolo et al., 2021). Since Fe biogeochemistry is strongly coupled with the global carbon cycle, this element receives considerable attention by the ice core community (Wolff et al., 2006) and methods have been developed to measure its concentration and speciation in ice cores (Spolaor et al., 2013; Conway et al., 2015). But until now, the effects of post-depositional processes on its geochemistry has been only poorly investigated.

The aim of the present work is to describe and discuss how climate and post-depositional processes affect dust in the Talos

Dome ice core (TALDICE, East Antarctica), with a focus on Fe speciation and mineralogy.

## 2    The TALDICE ice core

TALDICE has been drilled at Talos Dome (72°49′S, 159°11′E; 2315 m a.s.l.), a peripheral accumulation dome in the Ross Sea sector of East Antarctica (Frezzotti et al., 2004). It is 1620 m long and its ice age at the depth of 1438 m (*depth* refers to the distance between the ice surface and the considered section of the core) is ~150 ky BP according to the AICC2012 chronology

(Veres et al., 2013; Bazin et al., 2013), but there is evidence that the water stable isotope record is preserved up to ~1550 m deep, where the estimated ice age is ~343 ky BP, corresponding to Marine Isotopic Stage 10.1 (Crotti et al., 2020). Extensive research has been conducted on the dust record of TALDICE. Its concentration trend during the last climatic cycle reflects what observed at other East Antarctic sites, i.e. high atmospheric loads in glacial periods and lower ones during interglacials. A peculiarity of TALDICE is given by the influence of local Antarctic dust sources, corresponding to the ice-free sites of Victoria

Land region, close to Talos Dome (Delmonte et al., 2010; Baccolo et al., 2018b).

## 3    Materials and methods

### 3.1    Sample preparation

Fifty-four samples were prepared using 191 ice sections of TALDICE (each one 25 cm long). They consist in insoluble mineral particles extracted from meltwater and deposited on filtration membranes. The preparation took place in a clean room (ISO6)

installed at the University of Milano-Bicocca. Ice sections (~25x3x2 cm) were decontaminated with three baths in ultra-pure MilliQ water (Merck Millipore). They were stored in clean tubes under a ISO5 laminar flow bench until completed melting. Meltwater was split in two aliquots, one (~10 mL) for Coulter counter, the remnant for synchrotron radiation analysis.



The aliquots for Coulter counter, corresponding to single ice sections, were stored in plastic clean cuvettes (rinsed with MilliQ water) and added with a NaCl solution. The solution was prepared using MilliQ water and high purity solid NaCl
and before use it was filtered with 0.22 μm pore-size filters. This serves to make the liquid sample electrically conductive, a requisite for Coulter counter analyses. Final Na$^+$ concentration of samples was ~1% m/m. The aliquots for synchrotron radiation analysis were merged considering multiple ice sections (191 in total), so as to prepare 54 samples consisting each in a mass of suspended dust of at least 1 μg (estimated through Coulter counter). Merged samples were filtrated using hydrophilic PTFE membranes (ø 13 mm, pore-size 0.4 μm). Before filtration, membranes were rinsed for two weeks in a high purity
HNO$_3$ solution (concentration m/m 5%, weekly renewed). Filtration was operated with a micro-pipette, so as to concentrate the particles on the membrane in the smallest possible area (Macis et al., 2018). After filtration the membranes were placed in dedicated clean PTFE holders and sealed in plastic bags.

Samples were prepared considering the entire length of the core: 7 correspond to the Holocene (0-673 m, 0-11.7 ky BP), 7 to the last deglaciation (674-827 m, 11.7-18 ky BP), 5 to MIS2 (828-951 m, 18-30 ky BP), 15 to MIS3 (952-1259 m, 30-60 ky
BP), 3 to MIS4 (1260-1292 m, 60-80 ky BP), 6 to MIS5 (1293-1418 m, 80-146 ky BP), 1 from MIS6 (1419-1438 m, 146-154 ky BP) and 11 to the deep part of TALDICE not dated by AICC2012 chronology (1439-1620 m), but partially dated by the new TALDICE-deep chronology (Crotti et al., 2020).

### 3.2  Coulter counter

To determine the concentration and grain size of insoluble dust particles, the Coulter counter method was used. Samples were
measured with a Beckman Multisizer 4, equipped with a 30 μm orifice to measure the concentration of particles between 0.6 and 18 μm divided into 256 channels. Details are found in Delmonte et al. (2002).

### 3.3  Synchrotron radiation spectroscopic measurements

The application of synchrotron light to determine the elemental and mineralogical composition of TALDICE dust, was performed at beamtime B18 of the Diamond Light source (Cibin et al., 2019). A glovebox was constructed and connected to
the experimental chamber of the beamline to handle the samples in clean conditions. Many additional precautions were also adopted to limit contamination and increase the signal to noise ratio (Baccolo et al., 2018a).

### 3.3.1  X-ray fluorescence spectroscopy

Major elements (Na, Mg, Al, Si, K, Ca, Ti, Mn, Fe) in dust were investigated through synchrotron radiation X-ray fluorescence spectroscopy. Samples were irradiated with a 10 keV incident beam (cross section ~1x1 mm) for 600s and the fluorescence
signal was acquired with a silicon drift detector. Full details, including analytical performance, are given in Baccolo et al. (2018a, b). Elemental concentrations were converted into oxides concentrations and closed to 100%, following an established practice (Rudnick and Gao, 2003).





### 3.3.2 X-ray absorption spectroscopy

Speciation and mineralogy of the Fe fraction of TALDICE dust were investigated through X-ray absorption near edge structure
spectroscopy (XANES), performed at the Fe K-edge transition. For each sample three measurements were carried out, acquir-
ing the fluorescence signal of the samples at steps of 0.15 eV and considering the interval between 7000 and 7400 eV. Spectra
were calibrated, normalized and averaged using the Athena software (Ravel and Newville, 2005). Three spectral features were
gathered: 1- the energy of the Fe K-edge transition; 2- the energy of the pre-edge peak centroid; 3- the intensity of the pre-edge
peak; see Figure S1 for details. The energy of both pre-edge peak and Fe K-edge transition are directly related to the oxidation
state of Fe, while the intensity of the pre-edge peak depends on its coordination and symmetry considerations (Berry et al.,
2003).

### 3.3.3 Relative abundance of Fe-bearing minerals

Comparing dust samples with the ones corresponding to 14 Fe-mineral references (biotite, chlorite, glaucophane, goethite,
hematite, hornblende (ferro-honblende), jarosite, magnetite, muscovite, fayalite, pargasite, pyrite, schoerlite, siderite), it was
possible to estimate the contribution of single Fe-bearing minerals into the samples. XANES spectra of reference minerals
were collected following the same procedure adopted for TALDICE samples (Figure S2). XANES spectra of ice core dust
were reproduced through ordinary least square regression, using linear combinations constructed with mineral spectra (Figure
S3). For each sample all the combinations defined by 4, 3, 2 and 1 mineral references (1456 combinations per sample) were
calculated through ordinary least square regression and the best one (in terms of R-squared, it always exceeded 0.9) was
selected to represent the sample. In some cases the second best-fit combination had an R-squared close to the best-fit one, but
the difference between the two combinations always regarded the less abundant of the 4 selected references, with negligible
effects on data analysis and interpretation. The combinatoric package of the Athena software was used (Ravel and Newville,
2005). Following the procedure adopted by Shoenfelt et al. (2018) and Liu et al. (2018), the relative abundance of Fe-bearing
minerals was estimated considering the % linear coefficients of the selected combinations.

## 135  4   Results and discussion

### 4.1   The TALDICE dust record

Figure 1 shows the dust record of TALDICE. Considering the last climatic cycle (i.e. the Holocene and MIS 2-3-4), some well
known features characterizing the relationships between the atmospheric dust cycle and Antarctic climate are recognized. The
most evident is the negative correlation between dust concentration and $\delta^{18}O$ (1 m resolution Stenni et al. (2011)). In accordance
with the suppression of dust production and transport from remote sources during interglacials (Albani et al., 2012b), the mean
dust concentration in TALDICE Holocene ice is ~25 ng g$^{-1}$, while during MIS2, corresponding to the last glacial maximum,
it exceeds 300 ng g$^{-1}$, as a consequence of the massive activation of South American sources (Sugden et al., 2009) and to the
enhanced atmospheric transport toward Antarctica (Markle et al., 2018). The shift from interglacial to glacial conditions not





**Table 1.** Dust concentration values in TALDICE. For each climatic period mean concentrations are reported along with standard deviations.

| Period | Holocene | Degl. | MIS2 | MIS3 | MIS4 | MIS5 | MIS6 | MIS7 | MIS8-9 | deep part |
|---|---|---|---|---|---|---|---|---|---|---|
| (kyr BP) | 0-11.7 | 11.7-19 | 19-31 | 31-58 | 58-68 | 68-132 | 132-190 | 190-246 | 246-337 | 337-unk. |
| conc. (ng g$^{-1}$) | 26±18 | 104±125 | 317±174 | 80±31 | 116±59 | 61±55 | 158±190 | 182±302 | 121±73 | 110±123 |
| FPP (%) | 50±4 | 57±8 | 63±8 | 55±7 | 53±5 | 52±5 | 54±7 | 45±9 | 35±10 | 36±9 |
| CLPP (%) | 19±7 | 14±7 | 6.5±3.4 | 10.1±4.8 | 9.6±3.9 | 13±8 | 9.1±5.2 | 21±14 | 21±6 | 24±11 |

only affects dust concentration, but also its grain size, as revealed by the FPP (fine particle percentage) and CLPP (coarse

particle percentage) indexes. The first one is the relative concentration of particles between 0.6 and 2 μm with respect to the 0.6 - 5 μm interval; CLPP is the ratio between the concentration of particles between 5 and 10 μm and the total concentration of particles between 0.6 and 10 μm. During the Holocene, FPP has a mean value of 50 %, while during MIS2 it increases to 63 %, revealing that under glacial conditions dust particles deposited at Talos Dome are relatively smaller compared to interglacial periods. Similar evidence has already been reported from other East Antarctic sites and interpreted considering that

dust transported to East Antarctica during glacials is subject to long-range and high-altitude atmospheric pathways, allowing for the efficient removal of coarse particles (Delmonte et al., 2002). CLPP has a mean value of 19 % during the Holocene and of 6.5 % during MIS-2. This index is indicative of the relative abundance of particles larger than 5 μm, which, given the coarse diameter, are related to local Antarctic sources (Albani et al., 2012a; Baccolo et al., 2018b). The decrease of the index during glacial periods has to be interpreted in relative terms. It has been observed that the absolute activity of local Antarctic sources

is only partially influenced by climate (Baccolo et al., 2018b). CLPP is lower in glacials not because of a reduction of coarse particles, but because of an increase of the fine ones from South America.

MIS4 presents features similar to MIS2. During MIS5.5, the previous peak interglacial warmth, dust concentration reaches a low associated with high values of δ$^{18}$O and CLPP, compatible with a regionalization of the atmospheric dust cycle in the Talos Dome area, similarly to the Holocene. Going deeper the relationship between dust and climate is less evident. Looking

at Figure 1 and Table 1, it can be appreciated that below 1430 m, dust concentration exceeds 100 ng g$^{-1}$ with dampened oscillations. CLPP exceeds 20 % and FPP drops to values between 45 and 35 %, highlighting that in deep TALDICE dust particles are coarser. The increase of dust grain-size is interpreted as a consequence of aggregation, a post-depositional process occurring in deep ice (De Angelis et al., 2013).

Size distributions of insoluble particles also point to dust aggregation. Dust from the upper part of the core (Figure 2e-h),

presents a mode between 1 and 2.5 μm and a tail of particles larger than 5 μm; this suggests a mix from remote and local sources (Albani et al., 2012a). Dust from deep TALDICE (Figure 2j-l) is characterized by a higher abundance of coarse particles, a lack of fine ones and modal values exceeding 4 μm. Such features are not encountered in sections of Antarctic ice cores not affected by post-depositional alterations (Delmonte et al., 2002), they result from *in situ* aggregation of particles (De Angelis et al., 2013; Baccolo et al., 2021).

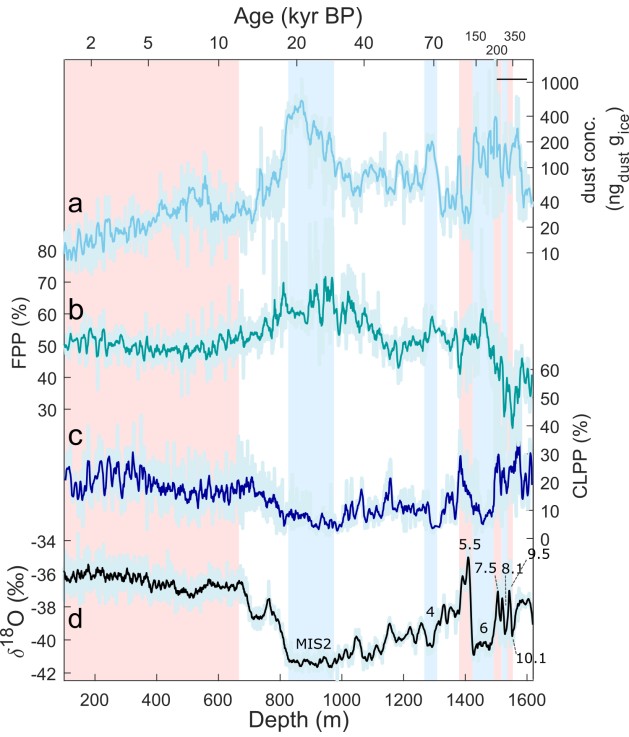

**Figure 1.** The mineral dust record of the Talos Dome ice core. From the upper to the lower curve: (a) total dust concentration of insoluble particles presenting a diameter between 0.6 and 10 μm (Baccolo et al., 2021); (b) FPP (fine particle percentage) ; (c) CLPP (coarse local particle percentage); (d) $\delta^{18}O$ (Stenni et al., 2011). Light blue bands highlight glacial culminations (marine isotopic stages 2-4-6-8), the red one the Holocene and marine isotopic stage 5.5.

Post-depositional processes in TALDICE also influence the climatic significance of its dust record. This is shown in Figure 2a-d, where the correlation between dust and ice stable isotopes is analyzed. The correlation is high during the last climatic cycle, but decreases in older periods. This is revealed by R-squared which decreases from 0.66 during the last cycle, to less than 0.3 in the deep core. The same conclusion is drawn looking at the trajectory describing the evolution of the $\delta^{18}O$ - dust pair. During the first cycle it well reproduces the transition from glacial to interglacial conditions and other events, such as the

Antarctic Cold Reversal and the partition of the last glacial period into MIS 2-3-4. This is partially true for the previous climatic cycle, but there the correlation decreases (R-squared from 0.66 goes to 0.49). The degradation continues in the deepest part of TALDICE, where the coefficient doesn't exceed 0.25 and the trajectories follow irregular paths, highlighting a substantial decoupling of dust concentration and isotopic signals. This is confirmed by a study (Crotti et al., 2020) showing that at Talos Dome below 1578 m deep, climatic signals are not preserved.





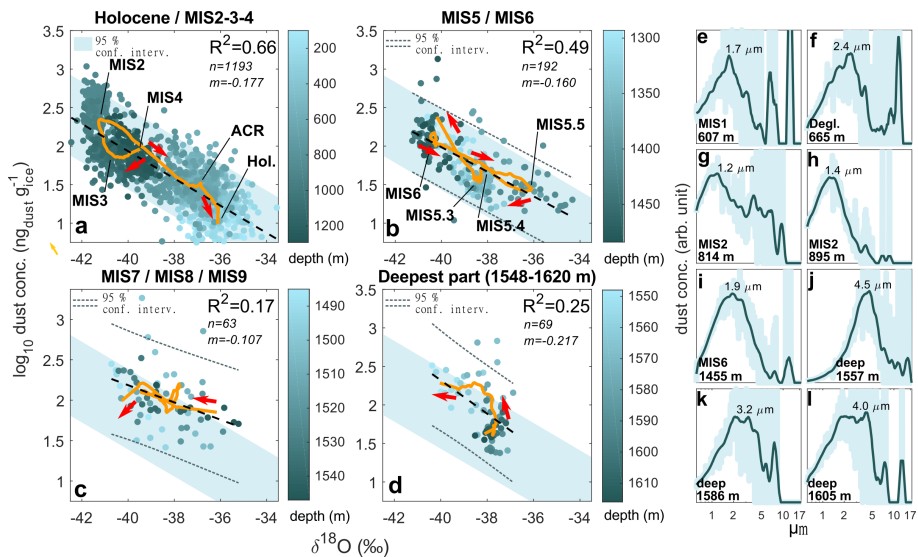

**Figure 2.** Panels a-d: linear correlation between $\delta^{18}$O and dust concentration (log) in the Talos Dome ice core. Blue bands correspond to the 95 % coinfidence level interval of the linear regression calculated considering the last climatic cycle; dashed lines highlight the coinfidence interval of each period. The orange curve is the trajectory showing the evolution of the $\delta^{18}$O - dust pair with time (red arrows). It was obtained applying a first order (40 % sample window) Savitzky-Golay filter to both variables. Panel e-l: dust size distributions from the Talos Dome ice core.

## 180  4.2   Major element composition of mineral dust

Comparing TALDICE dust with geochemical references, it is noticed that the best agreement is found with the Post Archean Australian Shale reference (PAAS) (Taylor and McLennan, 1985), not with the average Upper Continental Crust reference (UCC) (Rudnick and Gao, 2003). With respect to UCC, PAAS is depleted in mobile major oxides (such as CaO and Na$_2$O) and enriched in Fe- Ti- and Al-oxides. UCC refers to the whole upper continental crust, while PAAS is representative of surficial sedimentary rocks subject to chemical weathering (Taylor and McLennan, 1985). The last is richer in residual oxides while lacks labile soluble oxides. The similarity between TALDICE dust and PAAS is not unexpected, since atmospheric dust is produced at the Earth surface, where sedimentary and weathered rocks dominate.

Another feature emerging from Figure 3, is the change of dust composition along the core. Some oxides show significant variations with depth (see also Figure S4). The ones showing the most evident trends are SiO$_2$ (increasing) and CaO (decreasing), which pass from an average concentration of 64 % and 1.7 % in the Holocene to 74% and 0.4 % in the deep part of the core. Other oxides showing minor variations are: Na$_2$O (decreasing), MgO (decreasing), Al$_2$O$_3$ (decreasing) and K$_2$O (increasing). TiO$_2$, MnO and Fe$_2$O$_3$ are stable. These variations are not affected by climatic cycles, only by depth. This is an indication that they are likely related to post-depositional processes and not to primary changes of dust sources. The effect of disturbance from the bedrock must also be discarded since the ice stratigraphy at Talos Dome is uninterrupted until 1578 m deep (Crotti et al., 2020). Ca, Mg and Na, the elements showing the strongest decrease, are mobile and typically affected





**Table 2.** Average major element composition of TALDICE dust. Data are expressed as % mass fractions of oxides. For each climatic period mean values are reported with standard deviations.

| Period | Holocene | Degl. | MIS2 | MIS3 | MIS4 | MIS5 | MIS6 | deep part |
|---|---|---|---|---|---|---|---|---|
| (kyr BP) | 0-11.7 | 11.7-19 | 19-31 | 31-58 | 58-68 | 68-132 | 132-154 | 154-unk. |
| Na$_2$O (m/m %) | 3.1±1.1 | 2.1±1.0 | 1.9±0.5 | 2.0±0.5 | 4.3±3.3 | 2.3±1.0 | 0.2 | 1.6±0.9 |
| MgO (m/m %) | 1.6±0.4 | 2.0±1.0 | 3.0±1.6 | 0.8±0.3 | 0.8±1.1 | 0.4±0.3 | 0.5 | 0.6±0.8 |
| Al$_2$O$_3$ (m/m %) | 20.7±5.3 | 19.7±4.8 | 23±2.5 | 16.3±3.9 | 23.5±8.3 | 14.7±3.9 | 17.2 | 13.5±5.3 |
| SiO$_2$ (m/m %) | 63.9±5.0 | 65.9±6.3 | 62.9±3.4 | 72±3.6 | 63.9±4.5 | 73.4±3.9 | 72.9 | 74.2±5.6 |
| K$_2$O (m/m %) | 1.6±0.3 | 1.8±0.5 | 1.7±0.3 | 2.0±0.4 | 2.6±1.7 | 2.3±0.5 | 2.5 | 2.4±0.5 |
| CaO (m/m %) | 1.7±0.4 | 1.3±0.5 | 1.0±0.4 | 0.9±0.4 | 0.4±0.2 | 0.4±0.2 | 0.3 | 0.4±0.3 |
| TiO$_2$ (m/m %) | 1.5±0.4 | 1.6±1.3 | 0.9±0.2 | 0.9±0.2 | 0.4±0.2 | 0.9±0.3 | 0.8 | 1.0±0.2 |
| MnO (m/m %) | 0.06±0.02 | 0.05±0.01 | 0.06±0.02 | 0.04±0.02 | 0.05±0.04 | 0.04±0.01 | 0.04 | 0.04±0.02 |
| Fe$_2$O$_3$ (m/m %) | 5.8±0.6 | 5.6±1.3 | 5.4±1.1 | 5.1±0.8 | 4.0±0.1 | 5.5±0.6 | 5.6 | 6.3±1.5 |

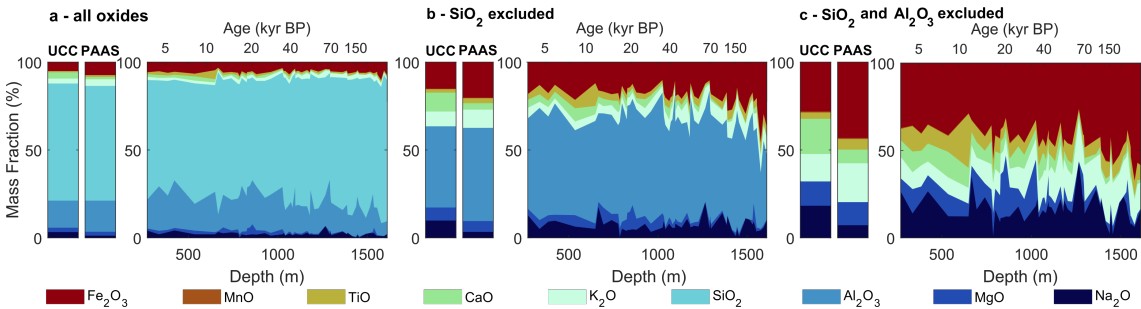

**Figure 3.** Major element in dust from the Talos Dome ice core. Data are shown considering all major element oxides (panel a), excluding SiO$_2$ (panel b) and SiO$_2$ and Al$_2$O$_3$ (panel c). UCC (upper continental crust, Rudnick and Gao (2003)) and PAAS (post archean Australian shale, **?**) are shown for comparison.

by chemical weathering (Nesbitt and Young, 1982). Their reduction suggests a progressive chemical weathering of dust in TALDICE. In particular, the deepest samples lack of MgO and CaO, pointing to carbonate dissolution. The reaction between acidic species and carbonates, with the consequent precipitation of gypsum and other soluble sulphates, is one of the most investigated post-depositional processes in deep ice (Ohno et al., 2006; Iizuka et al., 2008; Traversi et al., 2009; Eichler et al.,

2019). Our evidence confirms that this reaction also occurs at Talos Dome and for the first time they are detected in relation to the compositional changes of insoluble dust. The increase of SiO$_2$ is likely relative and reflects the progressive loss of labile species.


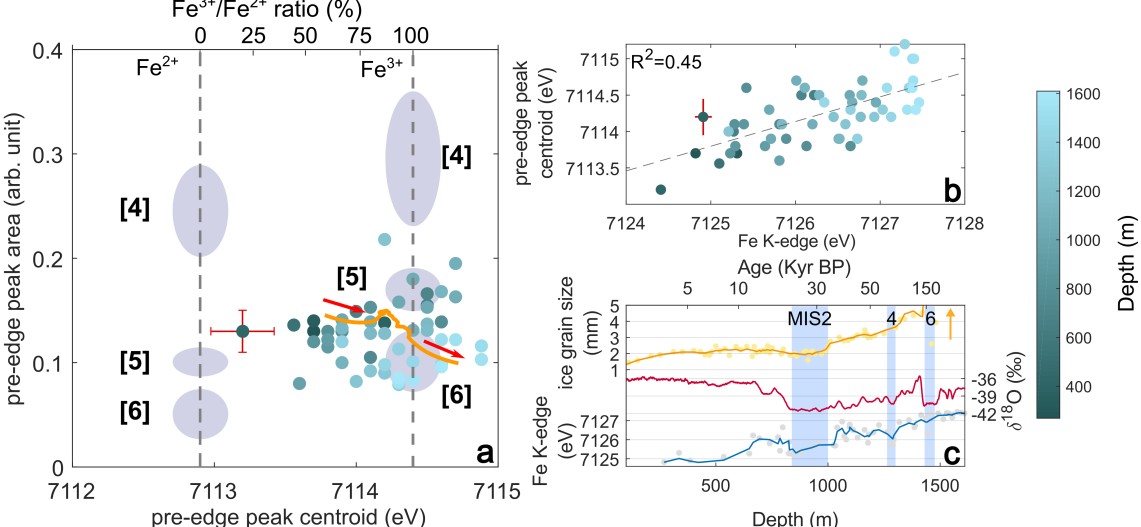

**Figure 4.** X-ray absorption near edge structure spectrosopy results from the analysis of dust in the Talos Dome ice core. Panel a: analysis of the Fe K- pre edge spectral region. The intensity and energy position of the pre-edge peak are shown following the scheme proposed by Wilke et al. (2001). Ellipses and numbers in brackets refer to the coordination of Fe, vertical lines to its two oxidation states; the orange curve represents the trajectory of the samples along the core, it was calculated as in Figure 2 (arrows from shallow to deep). Panel b: a comparison between the energy position of the pre-edge peak and the main K-edge transition (see Figure S1 for details). Panel c: the energy position of the Fe K-edge transition (blue line) vs. isotope composition of ice (red line, from Stenni et al. (2011)) and ice grain size (yellow line, from Montagnat et al. (2012), the arrow indicates the ice observed but not quantified ice crystals larger than 40 cm found below the depth of 1481 m); blue bands highlight MIS 2, 4 and 6. For each panel one point presents mean error bars (not visible in panel c because of scale).

### 4.3 Iron oxidation and coordination

The most evident result from XANES concerns the progressive oxidation of Fe with depth. This is appreciated through different
approaches in Figure 4. Figure 4a shows that the Fe fraction of dust in the first 1000 m of the core consists in a mixture of $Fe^{2+}$ and $Fe^{3+}$, reflecting the typical composition of mineral aerosol (Schroth et al., 2009). Deep samples show a pure $Fe^{3+}$ signature. With respect to coordination, samples display an octahedral symmetry (coordination number 6), with secondary inputs from other geometries. This is also in accordance with observations concerning aerosols (Wilke et al., 2001; Formenti et al., 2014).

Figure 4b and c focus on Fe oxidation. Panel b shows the correlation between the energy of the pre-edge peak and of the
K-edge transition, both correlated with Fe oxidation (Berry et al., 2003). Panel c shows the variation of the K-edge energy transition along the core. The ~2 eV increase confirms the oxidation of mineral dust in deep ice. The process is rather continuous, regardless of the climatic oscillations, suggesting that it is not climate-dependent but relates to post-depositional changes. This is confirmed by the similarity of the oxidation trend and the growth of ice grains with depth (Figure 4c). There are two exceptions: 1- in the deepest part of the core the Fe K-edge reaches a stable energy, pointing to a complete oxidation; 2- in
correspondence with MIS2 and 4 the trend shows two slowdowns, as if oxidation was inhibited.





**Table 3.** Average relative abundance of Fe-bearing minerals TALDICE dust. Data are expressed as % abundances

| Period | Holocene | Degl. | MIS2 | MIS3 | MIS4 | MIS5 | MIS6 | deep part |
|---|---|---|---|---|---|---|---|---|
| (kyr BP) | 0-11.7 | 11.7-19 | 19-31 | 31-58 | 58-68 | 68-132 | 132-154 | 154-unk. |
| Hornblende (%) | 19.2 | 12.3 | 19.1 | 2.6 | 0 | 4.1 | 8.4 | 0.7 |
| Muscovite (%) | 11.5 | 12.9 | 18.0 | 13.0 | 7.6 | 2.0 | 0 | 0 |
| Siderite (%) | 2.9 | 3.9 | 0.4 | 4.2 | 1.9 | 2.1 | 0 | 1.1 |
| Magnetite (%) | 4.8 | 3.8 | 0 | 0.8 | 35.2 | 5,6 | 0 | 0.9 |
| Goethite (%) | 30.8 | 56.0 | 37.4 | 60.3 | 30.8 | 45.5 | 31.7 | 42.9 |
| Hematite (%) | 15.7 | 8.3 | 20.6 | 2.6 | 0 | 0 | 0 | 0 |
| Pyrite (%) | 5.9 | 1.8 | 0.5 | 1.8 | 3.3 | 1.9 | 1.7 | 2.1 |
| Jarosite (%) | 0 | 0 | 0 | 10.5 | 14.3 | 38.8 | 58.1 | 50.3 |
| Others(%) | 9.1 | 1.0 | 4.0 | 4.1 | 7.0 | 0 | 0 | 2.0 |

During glacial culminations (MIS 2, 4 and 6) the dominant source of dust transported to Antarctica was Southern South America (Delmonte et al., 2004, 2010). One of the processes responsible for the increased dust emission during glacials, is glacial activity producing deflatable sediments (Sugden et al., 2009). Glacial sediments are geochemically fresh and thanks to the limited atmospheric exposure they are only partially oxidized (Shoenfelt et al., 2017). The increase of $Fe^{2+}$ during MIS 2

and 4 is related to the transport of glacial dust from South America (Spolaor et al., 2013). During these periods the amount of dust deposited at Talos Dome acts as a buffer, neutralizing the acidity of ice and consuming reactive species (Ohno et al., 2005, 2006; Eichler et al., 2019). Figure 4c shows that the growth of ice grains is also temporarily inhibited during MIS2 because of pinning on ice re-crystallization by the high concentration of insoluble particles (Durand et al., 2006). Something similar is visible in MIS4, while ice corresponding to MIS6 does not present neither an $Fe^{2+}$ recovery, nor a decrease of ice grain size

(Figure 4c), probably because in deep ancient ice *in situ* oxidation of Fe-minerals and ice metamorphism are too advanced. On the contrary, in interglacial ice atmospheric acidity is more available for post-depositional reactions favoring oxidation and weathering, also thanks to a more efficient ice re-crystallization (Iizuka et al., 2008; Eichler et al., 2019).

### 4.4 Iron mineralogy

Fe-mineralogy results are shown in Figure 5, Figure 6 and Table 3. Only minerals whose average relative abundance (rel. ab.)
exceeds 2 % have been considered to the aims of the discussion. A first inspection of both Figures 5 and 6 reveals that the Fe-mineral assemblage of TALDICE dust is not uniform with depth.


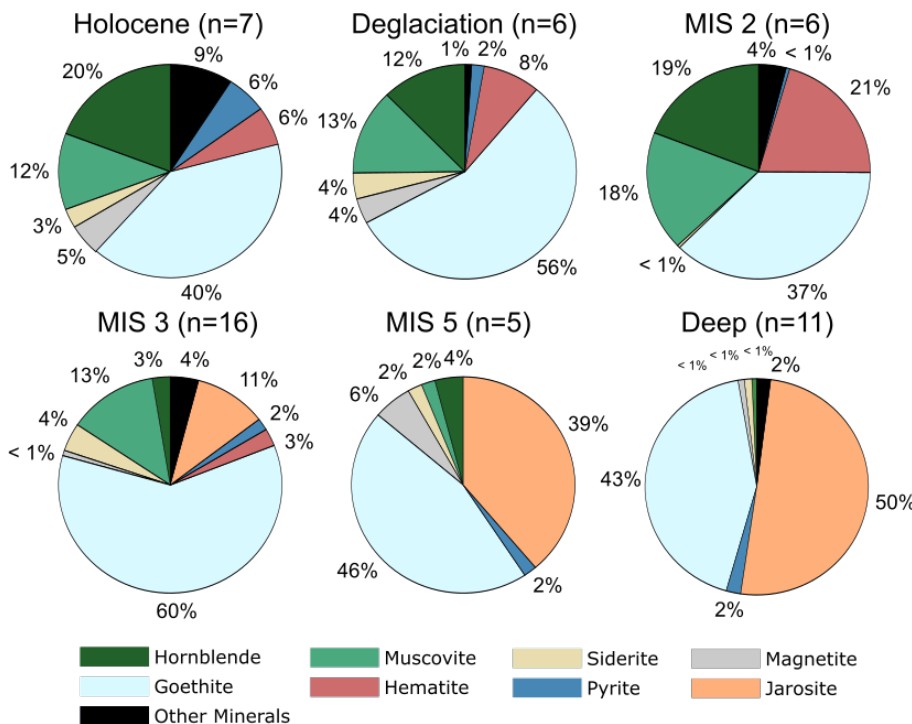

**Figure 5.** Pie-charts showing Fe-mineralogy of dust from the Talos Dome ice core. Each chart refers to a climatic period. Marine isotopic stages 4 and 6 have been excluded because of the low number of samples corresponding to these periods (2 and 1 respectively).

### 4.4.1 Hornblende and jarosite

The most evident trends regard hornblende and jarosite (Figure 6a and g). Since they involve the entire core regardless of climatic conditions, we interpret them as a consequence of post-depositional processes. Hornblende, after goethite, is the most

abundant Fe-mineral in Holocene (rel. ab. 19 %), but it rapidly decreases and in the deepest part of TALDICE it is almost absent (rel. ab. 0.7 %). Jarosite shows an opposite behavior. It is not present in the shallow part of TALDICE, it appears below 1000 m deep, becoming the dominant Fe-bearing mineral below 1400 m (rel. ab. 50 %). The two trends are linked. Hornblende is a common ferrous mineral present in South American dust (Shoenfelt et al., 2018) and in general is one of the dominant $Fe^{2+}$-bearing minerals in global aerosol (Schroth et al., 2009; He et al., 2020). On the contrary jarosite, a ferric sulphate, is not

common in atmospheric dust. Jarosite is well-known for being a weathering product and its widespread identification in deep TALDICE has been regarded as an evidence of weathering affecting dust in deep ice (Baccolo et al., 2021). The concurrent decrease of hornblende and increase of jarosite confirm Fe oxidation. Hornblende (and secondarily muscovite) seems the principal mineral whose dissolution leads to the consumption of $Fe^{2+}$, while jarosite build-up drives the accumulation of $Fe^{3+}$. The other minerals presenting a ferric component, with the exception of magnetite, also show a decreasing trend (muscovite,

siderite, pyrite) and are almost absent in the deepest part of the core.





### 4.4.2 Siderite and pyrite

Beside the trends involving the entire ice core, some minerals show an additional pattern in correspondence of MIS2. In some cases it is a relative maximum (muscovite, hematite), in others a minimum (siderite, pyrite). Given the correspondence with MIS2, such features are interpreted as climate-related signals. During MIS2 dust deposited on East Antarctica mostly comes

from Patagonia (Delmonte et al., 2004, 2010), while during the Holocene it is provided from the ice-free outcrops of Northern Victoria Land (Delmonte et al., 2010; Baccolo et al., 2018b). The shift of mineralogy between MIS2 and Holocene is related to this. Minerals which distinguish local Holocene dust from glacial Patagonian one, are siderite and pyrite. In the Holocene they constitute about 3 % each of the Fe-minerals in TALDICE, but during MIS2 they drop to 0.5 %. This agrees with the geology of Victoria Land, where they are relatively common owing to the basaltic/doleritic nature of local rocks (Sturm and

Carryer, 1970; Dow and Neall, 1974). In addition to the geologic context, also atmospheric transport could partially explain their absence during MIS2. Both minerals are easily oxidized when exposed to the atmosphere. Their lack in MIS2 could be related to their complete oxidation during the long-range transport from South America. It is known that mineral aerosol is subject to a number of chemical reactions, and Fe-oxidation is one of the most relevant (Shi et al., 2012). On the contrary, siderite and pyrite survive the atmospheric transport during the Holocene because of the proximity of local dust sources and

the short-range transport.

### 4.4.3 Muscovite

Dust deposited during MIS2 is rich in muscovite and hematite. Thanks to the aerodynamic shape of its crystals, muscovite is common in atmospheric dust subject to long-range transport (Engelbrecht and Derbyshire, 2010) and is one of the most abundant minerals deposited on East Antarctica during MIS2 (Delmonte et al., 2017; Paleari et al., 2019). The abundance of

this mineral in glacial ice is likely the main responsible for the slowdowns observed along the oxidation trend of Fe (Figure 4c). Fe in muscovite is in fact both ferric and ferrous and during MIS2 it is the only mineral, with hornblende, presenting a ferrous component, since siderite, magnetite and pyrite are almost completely absent (Figure 6). Something similar is observed in MIS4, when the second slowdown is observed, again corresponding with an increase of muscovite. In this second case both the slowing of oxidation and the increase of muscovite are less evident, probably because of the stronger influence of oxidation at

greater depth. Below 1300 m deep muscovite is completely absent, regardless of the climatic period, suggesting that also this mineral is affected by weathering in deep ice, such as hornblende. Being muscovite an Al-K silicate, its dissolution probably supplies a fraction of the K required for jarosite precipitation.

### 4.4.4 Hematite

Hematite (a Fe-oxide) is a weathering product in soils under dry and warm conditions, it is typical in tropical regions while it

is rarely encountered in cold and wet climates (Schwertmann, 1988). Along TALDICE it is mostly found during MIS2, when the dust signature is fully South American (Delmonte et al., 2010). During glacial culminations it has been proposed that an additional source other than Patagonia activates in South America, supplying dust to Antarctica. This is the Puna-Altiplano dry


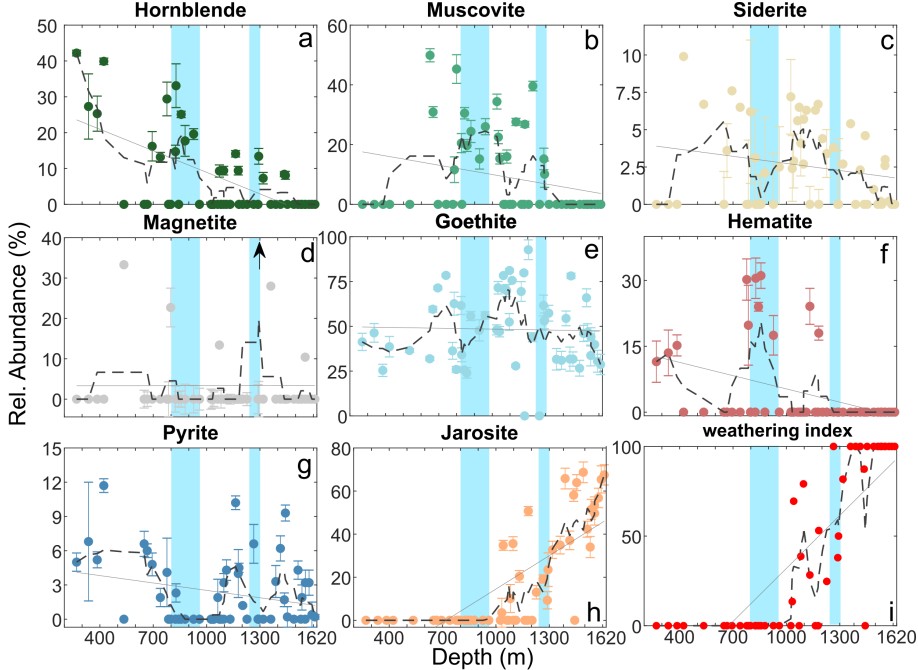

**Figure 6.** The abundance of the major Fe-bearing minerals identified in dust from the Talos Dome ice core with respect to depth. Dashed lines correspond to a 5-point moving average, solid grey lines to linear trends. In panel d one sample-point is out of scale (black arrow, magnetite relative abundance 70 %).

region, located in the tropical Andes (Delmonte et al., 2010), where hematite is widely present (Aubry et al., 1996). Our results show an excess of hematite during MIS2, supporting this hypothesis. A previous study focusing on inner East Antarctica

observed a higher abundance of hematite during the Holocene than in MIS2 (Paleari et al., 2019). The difference could be related to the geographic position of the two sites and to the relative influence of secondary sub-tropical sources during the Holocene. Below 1300 m deep, hematite is no more observed, suggesting that this mineral is not stable in deep ice. It is known that under acidic conditions (pH~4) hematite is not stable and can dissolve (Schwertmann and Murad, 1983; Zolotov and Mironenko, 2007), leading to the precipitation of jarosite and goethite (Papike et al., 2006).

**4.4.5 Goethite**

Considering the entire core, goethite is by far the dominant Fe-mineral in dust, with a mean rel. ab. of 46% (the second is jarosite, 18 % rel. ab.). The distribution of goethite with depth is rather uniform (Figure 6e). This is indicative of two things: 1- goethite is a common Fe-bearing phase in mineral aerosol (Formenti et al., 2014); 2-goethite is stable in the englacial environment, regardless of depth. The second point is corroborated by previous studies showing that at low temperature and

in acidic wet environments, goethite is the most stable Fe oxide-hydroxide (Schwertmann and Murad, 1983; Zolotov and Mironenko, 2007).



### 4.4.6   Magnetite

The mean rel. ab. of magnetite, a ferric-ferrous oxide, is about 3 % and does not show evident trends; only a single sample presents a high concentration (70.4 %). Without considering this anomalous value, possibly related to contamination or to the
presence of a micrometeorite (Rochette et al., 2008), mean value drops to 2 %. Despite its low rel. ab., magnetite is relatively stable along TALDICE and is not depleted in the deepest part, suggesting that it is not affected by oxidation-dissolution, as the other ferrous minerals considered in this study. This confirms its higher chemical resistance to acidic oxidation when compared to other Fe-minerals (Moncur et al., 2009).

### 4.5   Englacial weathering of dust

When considering the last climatic cycle -in particular the pair Holocene-MIS2 which corresponds to the first 950 m of the ice core- the chemical and physical properties of TALDICE dust are fully interpretable in the light of the well-known effects of major climatic swings on atmospheric dust. This is true for concentration, grain size and geochemistry, suggesting that the dust record is not significantly affected by post-depositional processes within this depth interval. Things change moving down: below 1000 m deep trends not related to climatic oscillations appear. One of the most evident is the presence of jarosite. The
mineral is first observed at 1000 m deep, and its concentration progressively increases toward the core bottom (Figure 5 and Figure 6h).

Jarosite is a Fe-K hydrated sulphate and results from chemical weathering (formula: $KFe_3(SO_4)_2(OH)_6$). It has never been reported in global mineral aerosol. These features, together with its increasing abundance, supports the hypothesis that jarosite is a product of englacial diagenesis related to chemical weathering of dust (Baccolo et al., 2021). Thanks to the conditions under
which jarosite precipitates, its formation returns information about the deep englacial environment and in particular about the occurrence of limited amounts of liquid water (ice pre-melting). Liquid water in deep ice has been extensively predicted by theories (Rempel et al., 2002; Ng, 2021) and in a few cases confirmed by observations (Fukazawa et al., 1998), but is still now a disputed argument (Eichler et al., 2019). The precipitation of jarosite is a strong indirect evidence for the presence of highly concentrated acidic liquid solutions in deep ice, since this ferric sulphate forms when limited amount of acidic (pH
< 4) aqueous solutions rich in solutes (also known as brines) interact with Fe-bearing minerals (Zolotov and Shock, 2005; Papike et al., 2006). At Talos Dome ice temperature at 1000 m deep, where jarosite appears, is -25.5°C (Rix & Martin personal communication). Combining this information with the phase diagram of the sulfuric acid-water pair (Beyer et al., 2003), it is possible to estimate that acidic brines forming at ice-grain junctions have a sulfuric acid concentration between 15 and 25 % m/m.

Re-crystallization and grain growth of deep ice are responsivle for the concentration of impurities. This is the only process that can promote the occurrence of such brines in deep ice through the local lowering of pressure melting point. This is supported by the similar trends of Fe oxidation and ice grain growth (Figure 4c). The acidity of deep brines is explained in the light of the strong incompatibility of acidic atmospheric species with respect to the ice molecular lattice (Wolff, 1996) and of their concentration at grain boundaries (Mulvaney et al., 1988), leading to the formation of acidic fluids.





The formation of jarosite not only alters the original mineralogical assemblage of dust in TALDICE, but can also be related to the anomalies in dust grain-size observed in deep dust (Figure 2e-l). Jarosite is well-known for creating a cementing matrix during weathering (Long et al., 1992). Its precipitation promotes the aggregation of mineral particles, altering the original granulometric signals. This has also been confirmed by microscopic observations (Baccolo et al., 2021). Considering all these elements, it is likely that chemical weathering also affects dust concentration, explaining the decrease of correlation with ice isotopic composition in the deep part of the ice core (Figure 2).

Jarosite formation is not the only dust-related anomaly in deep TALDICE. Another one regards the oxidation of Fe present in dust (Figure 4). The oxidation of Fe-minerals under acidic conditions is a well-known weathering pathway (Jones et al., 2014) which, among the others, leads to jarosite precipitation (Papike et al., 2006). Fe oxidation is also confirmed by the decline of many ferrous minerals in the deep part of the core. This is the case for hornblende, muscovite, siderite and pyrite (Figure 5, Figure 6 and Table 3). It is worth highlighting that the oxidation of pyrite, a process already known to occur in ice (Raiswell et al., 2018), supplies additional acidity to the englacial environment, further favoring jarosite precipitation.

An index was developed to summarize evidence from Fe-mineralogy: it is defined as the % ratio between the rel. ab. of jarosite and the sum between rel. ab. of jarosite, hornblende, muscovite and hematite; it is shown in Figure 6i. The index varies between 0 % in the upper part of TALDICE (pristine mineral assemblage) and rapidly increases to 100 % between 1000 and 1300 m, reflecting dust weathering.

The trend of some elements in the deep part of TALDICE (Table 2, Figure 3) agrees with the scenario described above. The decrease of Ca, Na and Mg with depth is interpreted as an effect of acidic weathering. In presence of acidic aqueous solutions, mobile and soluble elements are easily mobilized (Nesbitt and Young, 1982). It is worth mentioning that Al oxide is also affected by depletion in deep TALDICE. This suggests that not only the labile and soluble fraction of dust undergoes weathering but also more stable fractions such as aluminosilicates. This is confirmed by the identification in deep Antarctic ice of secondary aluminum-sulphate (Ohno et al., 2014). To further investigate such aspects, a comparison between soluble and insoluble impurities in deep TALDICE would be desirable. Currently data about ionic records from TALDICE are available, but they concern the upper part of the core, not the deep one (Schüpbach et al., 2013; Mezgec et al., 2017).

TALDICE presents a number of peculiarities if compared to other East Antarctic ice cores. They can partially explain why post-depositional processes affecting dust are so notable at this site. Dust deposited at Talos Dome has a local basaltic-doleritic signature (Baccolo et al., 2018a) and basalts are weatherable rocks, also at low temperature (Li et al., 2016; Niles et al., 2017). In addition, Talos Dome is located near the Southern Ocean and receives a considerable amount of marine aerosol rich in reactive acidic species (Iizuka et al., 2013). A further factor to consider is ice temperature. Talos Dome is near the coast and its climate is tempered by the ocean. When considering the same depth, ice temperature at Talos Dome is about 15-20 °C warmer than at inner sites (Talalay et al., 2020).

All these local features are likely related to the ice-metamorphism observed in deep TALDICE (Montagnat et al., 2012), where ice crystals up to 40-50 cm have been observed below 1480 m. High ice-temperature and metamorphism could partially explain why dust alteration is so relevant in TALDICE, while at inner sites the original properties of dust seem preserved at greater depth and further back in time (Delmonte et al., 2004; Kawamura et al., 2017). Replicating this study at Inner Antarctic


sites will be essential to distinguish the processes depending on the local characteristics of single sites from the ones more deeply related to ice depth and age.

## 5 Conclusions and perspectives

This study provides a first description of dust chemical weathering in deep polar ice. Dust grain size, concentration, mineralogy and composition are all affected by post-depositional processes. Fe speciation and mineralogy are efficient probes to explore
such processes in Antarctic ice. The englacial precipitation of jarosite, the concurrent decline of ferrous minerals and the depletion of some major elements suggest that below 1000 m deep, TALDICE is affected by acidic-oxidative weathering resulting from the interaction of dust with highly saline and acidic brines. From this perspective deep Antarctic ice can be seen as a "geochemical reactor", capable of promoting the precipitation of secondary minerals and the concurrent dissolution of others.

These findings pose issues to the interpretation of future deep Antarctic ice cores. Because of diffusive processes, the isotopic signal of ice, as well as other proxies (Barnes et al., 2003), is progressively deteriorated with depth (Jones et al., 2017), making difficult to isolate the climatic content of deep ice cores. Being considered relatively immobile and stable, dust has been regarded as a candidate to overcome these difficulties. This study demonstrates that also dust-related signals are touched by post-depositional transformations which need to be addressed.

It would be desirable to replicate the present study considering deep ice cores from inner East Antarctica where it would be possible to unveil additional processes related to the conditions found at 2000-3000 m ice deep. Another future implementation will be the concurrent analysis of soluble and insoluble impurities, including elements other than Fe. This will help to investigate the dissolution and precipitation of primary and secondary minerals and identify the geochemical reactions behind these transformations.

*Data availability.* Data used to the aims of the present study are available in the Supplementary Material. Full XANES spectra are found in the PANGAEA open repository, DOI: 10.1594/PANGAEA.924114

*Author contributions.* GB conceived the idea of this work. GB, BD, EDS prepared the samples and performed Coulter counter analyses. GB, GC, DH, AM carried out X-ray absorption and fluorescence analyses. GB interpreted the data and wrote the manuscript with contributions from all the coauthors.

*Competing interests.* The authors declare no competing interests.



*Acknowledgements.* This work is part of the TALDEEP project funded by MIUR (PNRA18-00098). The Talos Dome Ice core Project (TALDICE), a joint European programme, is funded by national contributions from Italy, France, Germany, Switzerland and the United Kingdom. Primary logistical support was provided by PNRA at Talos Dome. This is TALDICE publication no XX. This publication was generated in the frame of Beyond EPICA. The project has received funding from the European Union's Horizon 2020 research and innova-

tion programme under grant agreement No. 815384 (Oldest Ice Core). It is supported by national partners and funding agencies in Belgium, Denmark, France, Germany, Italy, Norway, Sweden, Switzerland, The Netherlands and the United Kingdom. Logistic support is mainly provided by PNRA and IPEV through the Concordia Station system. The opinions expressed and arguments employed herein do not necessarily reflect the official views of the European Union funding agency or other national funding bodies. This is Beyond EPICA publication number xx. The authors acknowledge Diamond Light Source for provision of beamtime within proposals sp7314, sp8372 and sp9050. We thanks

Paolo Gentile for providing mineral standards and also Paul Niles and Tanya Peretyazhko for the fruitful discussions



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
