# Peer review of "Deep ice as a geochemical reactor: insights from iron speciation and mineralogy of dust in the Talos Dome ice core (East Antarctica)"

_The Cryosphere, 2021_

## Referee Comment (RC1)

**Review of:** Deep ice as a geochemical reactor: insights from iron speciation and mineralogy of dust in the Talos Dome ice core (East Antarctica)
**Submitted to:** The Cryosphere Discussions
**Reviewer:** Nicolas Stoll
* * *
**General comments:**

The manuscript tackles the important question of chemical reactions occurring in deep polar ice and the discussed, but so far not observed, option of ice as a chemical reactor. Impurities in polar ice are of importance for several reasons, but dedicated studies on the processes taking place in the ice are rare. Thus, this manuscript is of interest for the cryo-community and offers new exciting results worthy of publication. The current "re-birth" of detailed impurity studies with new or improved techniques (e.g., Synchrotron radiation spectroscopy, LA-ICP-MS) will hopefully help to tackle the challenges ahead regarding Beyond EPICA Oldest Ice.

The presented manuscript, figures and data are of good quality and I mainly have suggestions regarding details. Unfortunately, there are several small issues regarding language, I would thus recommend a dedicated language check. However, the manuscript content is of high quality and I recommend it for publication with minor revisions.

Specific comments:

1) TALDICE should be explained earlier in the introduction or methods. The varying use of "TALDICE" or "Talos Dome ice" can be confusing, I would suggest to stay consistence in wording and to check for the proper use. Throughout the text TALDICE sometimes misses specific affixes such as "TALDICE ice" or "TALDICE dust", the acronym describes the entire ice core and not the described section or parameter in detail. Adding TALDICE ice or TALDICE dust would be enough to increase readability. Furthermore, I suggest to merge the (very short) section on TALDICE with the Materials and methods section.

2) P. 2 l. 24 "growing numbers of studies", please give some examples.

3) In general, the methods are very short and largely refer to other publications. This is a neat way to minimize word count, but 1-2 more sentences for each method, explaining the basics, would enable the broad audience of *The Cryosphere* to follow the manuscript directly. Especially more details on Synchrotron radiation spectroscopic measurements and XANES will be useful since it is relatively new to the cryo-community. One or two examples of the "many additional precautions" taken would be helpful on P. 4 l. 105.

4) P. 4 methods: It is described that 54 samples were prepared, the samples per climate period however add up to 55 (l. 93ff). What is the correct number of samples or did I miss something?

5) A table (possibly in the appendix) providing the formulas of the minerals introduced in 3.3.3. would enhance readability and make it easier to follow the interpretations and weathering effects etc. (4.4.1-4.5). It could even make sense to display the

chemical formula of the weathering processes resulting in jarosite, but I leave that to the authors.

6) P. 6 l. 162: I agree that the observed increase in dust size can be due to aggregation. However, "deep ice" is always relative and it would be good to mention that De Angelis et al. (2013) studied much deeper (~2900 m) and older (>400 ky) ice. Furthermore, Simoes et al. (2002) described something similar for Vostok (3450-3540 m / > 400ky).

7) P. 6 l 168 ff. It´s a very strong statement and other references for other Antarctic ice cores might be needed in addition to Delmonte et al. (2002). Additionally, it is difficult to argue that no post-depositional alteration occurred in EPICA ice since this was not the main objective of Delmonte et al. (2002) (Maybe I missed this explicit statement in the paper though).
I suggest to mitigate and rephrase the sentence to something like "high abundance of coarse particles is probably a result from in situ…"

8) P. 8 l. 181f link Fig. 3 here.

9) P. 9 l. 200: Define what is detected for the first time.

10) Check wording for "concern", "appreciate", xxxx m deep

11) P. 11 l. 221f The cited papers did not analyse Talos Dome ice and it should be stated clearly that only the processes are addressed here. You could add "…reactive species as proposed for Dome Fuji (Ohno et al., 2005) and EDML (Eichler et al., 2019)." Furthermore, the modelled pinning process by Durand et al, 2006 hasn't been observed yet in polar ice (e.g., Faria et al, 2010, Eichler et al, 2017, Stoll et al, 2021). Pinning is likely to play a role in the grain size evolution, but without a dedicated study on the microstructure of MIS 2 from TALDICE we can´t say for sure. Rephrasing to e.g., "probably related to pinning of grain boundaries by insoluble particles as suggested by Durand et al 2006." Could solve this issue.
What do you mean with "ice metamorphism is too advanced"? Do you refer to the dynamic recrystallization at 1141 m mentioned in Fig. 2 from Montagnat et al., (2012)?

12) I suggest to go through the structure of 4.4. and 4.5 again, the mixture of results and discussion is sometimes difficult to follow. Ways to structure it are e.g., from most abundant (absolute ) to least abundant or relative occurrence with depth. Discussing one mineral per subsection and always starting in the same way could enhance readability, e.g., either first presenting the original origin/process leading to the formation of the mineral or describing your data i.e. how much you observed (I would start with your data first and then discuss the origins and other issues).

I would briefly describe Jarosite in 4.4.X (abundance with depth) which can then lead to the "weathering" subsection explaining the origin. I think this link is one of the main strengths of the manuscript and should thus be presented clearly. I support the idea of the weathering index, but would describe it earlier in the text and also briefly in the figure caption (see comment on Fig. 6).
Jarosite is not found in the 3 youngest climate periods displayed in Fig. 5, so please restructure the sentence in P.12 l 233. (e.g., Hornblende and Jarosite are found throughout large parts of the core, Hornblende dominates the young/shallow

samples while Jarosite becomes more dominant with depth.)

4.4.2 starts with several sentences which could also be part of the introduction of 4.4. It would be easier to follow by keeping the structure, i.e. first stating the results and then discuss them. Are the 3% pyrite and siderite really "relatively common"?

13) Some statements are made and the used reference are mentioned a sentence or two later (e.g., P. 13 l. 275 "..it has been proposed that additional…"). Naming your references first or connecting the sentences would back up your arguments better.

14) On p. 14 l. 282 it is stated that hematite is not stable in deep ice. However, Hematite was observed e.g., at 2390 m (transition MIS 5.1 to 6) by Eichler et al. (2019) with Raman spectroscopy. To clarify this issue you could simply add "in our analysed TALDICE ice samples" and briefly describe other findings of hematite in deep ice. To explore the processes in deep ice further it would be interesting to know if there is a pH record for TALDICE which could back-up the hypothesis of acidic conditions?

15) Think about adding Kuiper et al. (2020b) to the section on pre-melting.

16) P. 15 l. 320ff. To my knowledge the processes acting on impurities, microstructure and grain growth are not fully understood yet (e.g., Eichler et al. 2019, Stoll et al., 2021). There might be more processes in the ice resulting in the concentration of impurities in addition to large deposition events as suggested for cloudy bands. Which process are you addressing here?

17) P. 16 l. 328 I suggest to explicitly link Fig. 2 from Baccolo et al. (2021) to the microscopic observations.

18) P. 16 Give some specific examples if possible, e.g., l. 341 "The trend of some elements…", l. 349 "TALDICE presents a number of peculiarities, such as…, if compared to…". Later you can explain them in detail, but a brief overview helps to follow.

19) I know it is very difficult to define (and to prove) but I would be interested in your thoughts on the exact processes involving high ice-temperature and ice metamorphism altering the dust record. This goes beyond the scope of this study, but is somehow crucial to understand (deep) ice better.

20) Sect. 5 Conclusions and perspectives:

The results of this study shine new light on the area of impurities in polar ice. Thus, I suggest so include more details regarding the results/discussion in the conclusions and to shift the perspectives part to a dedicated subsection of the discussion (e.g., which other elements/minerals would be interesting to investigate deep ice as a geochemical reactor).

References:

Delmonte, B., Petit, J., and Maggi, V.: Glacial to Holocene implications of the new 27000-year dust record from the EPICA Dome C (East Antarctica) ice core, Climate Dynamics, 18, 647–660, 2002.
de Angelis, M., Tison, J. L., Morel-Fourcade, M. C., and Susini, J. (2013). Micro-investigation of EPICA Dome C bottom ice: evidence of long term in situ processes involving acid-salt interactions, mineral dust, and organic matter. *Quat. Sci. Rev.* **78**, 248–265. doi:10.1016/j.quascirev.2013.08.012
Simoes, J.C., Petit, J.-R., Soichez, R., Lipenkov, V. Y., de Angelis, M., Liu, L., Jouzel, J., Duval,

P., 2002. Evidence of glacial flour in the deepest 89 m oft he Vostok ice core. *Ann. Glaciol.* 35, 340-346

Eichler, J., Weikusat, C., Wegner, A., Twarloh, B., Behrens, M., Fischer, H., et al. (2019). Impurity Analysis and Microstructure Along the Climatic Transition From MIS 6 Into 5e in the EDML Ice Core Using Cryo-Raman Microscopy. *Front. Earth Sci.* **7**, 1–16. doi:10.3389/feart.2019.00020

Kuiper, E.-J. N., de Bresser, J. H. P., Drury, M. R., Eichler, J., Pennock, G. M., and Weikusat, I. (2020a). Using a composite flow law to model deformation in the NEEM deep ice core, Greenland – Part 2: the role of grain size and premelting on ice deformation at high homologous temperature. *Cryosphere* **14**, 2449–2467. doi:10.5194/tc-14-2449-2020

Stoll, N, Eichler, J., Hörhold, M., Shigeyama, W. and Weikusat, I. (2021) A Review of the Microstructural Location of Impurities in Polar Ice and Their Impacts on Deformation. *Front. Earth Sci.,* doi: 10.3389/feart.2020.615613

Baccolo, G., Delmonte, B., Niles, P.B. *et al.* Jarosite formation in deep Antarctic ice provides a window into acidic, water-limited weathering on Mars. *Nat Commun* **12,** 436 (2021). https://doi.org/10.1038/s41467-020-20705-z

Figures

Figure 1 caption: The red bands highlight the Holocene and marine isotopic stages 5.5, 7.5 and 9.5.

Figure 2: The plots are rather busy and the use of the Holocene 95% confidence interval in 2b-d is not trivial to understand with the current legend. It might help to have the same legend in each panel and highlighting that the Holocene is always the reference.
Caption: Consider adding (arrows from shallow to deep) for the trajectories as done in Fig. 4, "with time" can be interpreted both ways.
Out of curiosity: why the jump in displaying the dust size distribution from MIS2 to MIS6?

Figure 3 caption: PAAS reference seems to be missing.

Figure 4: Minor, but the depth axis is the other way around compared to other figures. Having it always the same way would enhance understandability.

Figure 6: Please describe the weathering index briefly in the caption, for more details you can then refer to the section in the main text.

Technical corrections:

P. 1 l. 12 "allowed" change to "allow"

P. 2 l. 22 to clarify between dust and physical properties of ice consider changing "grain size" to "dust grain size"

P. 2 l.  26 sentence is difficult to understand, please rephrase "the older the ice"

P. 2 l. 30 a brief definition of "metamorphism of ice" would be good

P. 2 l. 35 pressure, stress and strain-rate

P. 2 l. the bottom of polar ice cores, true for Greenland too

P. 2 l. 38 check sentence, "…stable and its concentration"

P. 2 l. 39 "untouched" consider using "unaltered" throughout the manuscript

P. 2 l.41 consider restructuring the sentence

P. 2 l. 50 "to take place in ice are"

P. 3 l. 53 define very deep englacial environments

P. 3 l. 56 "secondary Fe-bearing minerals", maybe give 1 or 2 examples

P. 3 l. 58 first use of TALDICE – please define acronym here

P. 3 l. 61 "considerable attention" might need more references than Wolff et al., 2006

P. 3 l. 62 "the effects of… have been only…"

P. 3 l. 64 define your objectives with some more details

P. 3 section 2 consider merging with Materials and methods

Own paragraph on dust in TALDICE, define studies of "extensive research"

p.3 l. 72 "…what was observed at…"

P. 3 l. 74 sites of the Victoria Land region

P. 3 l. 78 consider rephrasing for clarity: 191 25 cm long section of TALDICE ice. Or in brackets (25 x 3 x 2 cm)
 They consist of insoluble mineral…

P. 3 l. 81 "until completely melted" or "melting was completed"

p.4 l. 90 Consider changing the sentence to "Filtration was done with a micro-pipette to concentrate the particles on the membrane in the smallest possible area."

P. 4 l. 109 "600 s"

P. 4 l. 111 "…100% following an…"

P. 5 l. 138 "Antarctic climate visible"

P. 5 l 139 please refer again to figure 1 for the dust concentration (Stenni et al., 2011)
P. 5 l 144 Consider a new paragraph on dust size

P. 6 l. 144 consider spelling FPP and CLPP in full first and then in brackets

P. 6 l. 148 "relatively small compared to"

P. 6 l 149f give examples and references for other sites
…allowing the efficient removal of coarse particles".

P. 6 l. 160 consider changing "appreciated" to "it is visible/ Fig. 1 and Table 1 show that below"

P.7 l. 170 "in TALDICE ice"

P. 8 l. 185 "The latter is richer in residual oxides while lacking labile soluble oxides.

P. 8 l. 188ff "from Fig. 3 is the change…", deep part of the core, respectively. Also check for consistency of the spacing of "%".

P. 9 l. 197 "…the deepest sample lack MGO and CaO, indicating carbonate dissolution,…"

Fig. 4 caption: "the arrow indicates the observed but not quantified ice crystals…"

P. 10 subsection 4.3 I suggest to extent the title to "coordination symmetry"

P. 10 l. 105 "…core consists of a mixture of…"

P. 10 l. 213 "…confirmed by the similarity of the oxidation trend…" The overall pattern is comparable, but consider rephrasing. Also, grain growth is impacted by several processes (impurities, recrystallization, temperature) and the sentence should be rephrased.

P. 11. l. 217 "…during glacials is…."

P. 11 l. 218 "…fresh and only partially oxidized due to limited atmospheric exposure"

P. 11 l. 230 "…have been considered in the discussion."

P. 12 l. 235 "Fe-mineral in Holocene ice…"

P. 12 l. 241 "…regarded as evidence of…"

P.12 l. 242 "…seems to be the principal mineral"

P. 13 l. 252 "Minerals, which… Patagonian dust, are… or "Minerals enabling to distinguish between…"

P. 13 l. 259f. Rephrase to "In the Holocene siderite and pyrite did not undergo chemical reactions during transport because of proximal dust sources are

P. 13 l. 265 "…main reason…"

P. 13 l. 270 Muscovite is completely absent below 1300 m indicating that this mineral is affected by weathering in deep ice, similar to hornblende. Muscovite, an Al-K silicate, probably supplies a fraction of the K required for…"

P. 15 l. 295 "…the mean value…"

P. 15 l. 308 "These features, …,support…"

P. 15 l. 320 "…of deep ice are responsible for"

P. 15 l. 321 "…local lowering of the pressure melting point."

P. 15 l.326 "…observed in deep ice…"?

P. 16 l. 331 "Another anomaly is the oxidation…"

P. 16 l. 333 "…which, among others, leads to…"

P. 17 l. 371ff "…making it difficult to isolate…". This stidy demonstrated that also dust-related signals are altered/impacted by post-depositional transformations.

*The Cryosphere* guidelines: The abbreviation "Fig." should be used when it appears in running text and should be followed by a number unless it comes at the beginning of a sentence, e.g.: "The results are depicted in Fig. 5. Figure 9 reveals that…".

---

## Referee Comment (RC2)

**Review of manuscript tc-2021-162: "Deep ice as a geochemical reactor: insights from iron speciation and mineralogy of dust in the Talos Dome ice core (East Antarctica)" by Giovanni Baccolo et al.**

25 August 2021

**Summary**

The authors present a new perspective of possible post-depositional processes that affect mineral dust records, particularly below 1000 m depth in the 1620 m long TALDICE ice core. The datasets are of high quality and obtained using Coulter counter and spectroscopic measurements such as Synchrotron radiation, X-ray fluorescence and XANES. Crucial properties of dust such as the concentration, grain-size, their elemental composition with a focus on Fe-mineralogy are discussed in depth. The study shows englacial formation of specific minerals in deep ice affecting the original scenario of dust deposition and conclude highlighting potential impacts while interpreting dust records on deep ice cores. While the originality, scientific quality and significance of the work is excellent, the manuscript falls short on language with several grammatical errors which needs language editing. I recommend this manuscript, after necessary language editing, for publication in the journal's special issue: Oldest ice: finding and interpreting climate proxies in ice older than 700000 years.

**Specific comments**

I do not mention any corrections regarding language as the manuscript needs thorough language editing. Following are some specific comments.

Line 35: The authors mention the role of depth and pressure in the post-depositional processes that has not been previously addressed - however this aspect of depth and pressure altering dust records aren't discussed in the results and discussion. I suggest to modify or

delete this sentence.

Line 38: replace ine with ice.

Section 2: Though co-ordinates are provided, I suggest a location map especially with surrounding dust sources would be useful for many.

Sample Preparation: I have some queries on technical aspects of sample preparation. You mention that the preparation took place in the ISO6 clean room - were the ice sections decontaminated under the laminar flow bench or in the clean room? At what temperature did this process take place? Also, considering the 2 cm thin ice sections used in this analysis, how thin was the ice after 3 baths decontamination? do you decontaminate the ice sections using ice cold ultra-pure water bath to avoid melting the sections that are already 2 cm thin?

I understand that you analysed 191 coulter counter samples and 54 filtered samples for spectroscopy. If not, you might have to clarify it in the sample preparation section.

line 78: remove extra "in".

Line 93: According to this paragraph, there are 55 samples, while you mention 54 in Lines 78 and 87. Your dataset in the supplement seems to have 54 samples.

Line 144: CLPP (coarse local particle percentage).

Lines 197–201: This paper focuses on many possible reactions of Fe-minerals happening in deep ice. The authors do mention about carbonate dissolution in deepest samples backed with well-known ice core studies. However, there is also a possibility that such post-deposition processes alter dust chemistry immediately after the snow deposition as shown from the surface snow cores by Mahalinganathan and Thamban (2016) that has not been observed in the holocene / interglacials of deep ice cores. Do you think the carbon dissolution and Fe-mineral reactions which are apparent in deep sections of TALDICE may be happening constantly from the time after snow deposition (instead of happening at a deeper section, (unless it is depth-pressure based), but are missed due to lesser spatial study?

Figure 1: $ng_{dust}g_{ice}^{-1}$ of dust concentration in figure. The caption misses mentioning MIS 7.5 and 9.5 for red bands.

Figure 3: Reference is not linked.

Figure 6: Choose contrasting colors for panels c, d and e.

Table 2: SD for MIS-6 column is missing.

**References**

Mahalinganathan, K. and Thamban, M.: Potential genesis and implications of calcium nitrate in Antarctic snow, The Cryosphere, 10, 825–836, 2016.

---

## Author Comment (AC1)

Replies to reviewers:

Reviewer #1

**Rev1:** The manuscript tackles the important question of chemical reactions occurring in deep polar ice and the discussed, but so far not observed, option of ice as a chemical reactor. Impurities in polar ice are of importance for several reasons, but dedicated studies on the processes taking place in the ice are rare. Thus, this manuscript is of interest for the cryo-community and offers new exciting results worthy of publication. The current "re-birth" of detailed impurity studies with new or improved techniques (e.g., Synchrotron radiation spectroscopy, LA-ICP-MS) will hopefully help to tackle the challenges ahead regarding Beyond EPICA Oldest Ice.

The presented manuscript, figures and data are of good quality and I mainly have suggestions regarding details. Unfortunately, there are several small issues regarding language, I would thus recommend a dedicated language check. However, the manuscript content is of high quality and I recommend it for publication with minor revisions.

**Reply:** We deeply thank the reviewer for his good acknowledgment of our work. We agree that in the manuscript several mistakes were present, we did our best to fix them improving the language.

**Rev1:** TALDICE should be explained earlier in the introduction or methods. The varying use of "TALDICE" or "Talos Dome ice" can be confusing, I would suggest to stay consistence in wording and to check for the proper use. Throughout the text TALDICE sometimes misses specific affixes such as "TALDICE ice" or "TALDICE dust", the acronym describes the entire ice core and not the described section or parameter in detail. Adding TALDICE ice or TALDICE dust would be enough to increase readability. Furthermore, I suggest to merge the (very short) section on TALDICE with the Materials and methods section.

**Reply:** We agree with the reviewer that our use of the acronym TALDICE was a little bit confusing. We have now introduced into already in the abstract and decided to use TALDICE when we refer to the ice core, while we talk about Talos Dome when we pose our attention on the site and not on the ice core. We have corrected the manuscript accordingly. We have also moved the section dedicated to the description of the core in the "Materials and methods" section.

**Rev1:** P. 2 l. 24 "growing numbers of studies", please give some examples.

**Reply:** We have now added the following references: "*Barnes et al., 2003; De Angelis et al., 2013; Ohno et al., 2016; Eichler et al., 2019; Baccolo et al., 2021*"

**Rev1:** In general, the methods are very short and largely refer to other publications. This is a neat way to minimize word count, but 1-2 more sentences for each method, explaining the basics, would enable the broad audience of The Cryosphere to follow the manuscript directly. Especially more details on Synchrotron radiation spectroscopic measurements and XANES will be useful since it is relatively new to the cryo-community. One or two examples of the "many additional precautions" taken would be helpful on P. 4 l. 105.

**Reply:** We tried to keep this part as much short as possible to limit the length of the manuscript, but we agree that the typical reader of the Cryosphere could be not aware about these techniques. We have added some passages:

- For Coulter counter: "*It relates small changes in the electrical conductance of meltwater (added with a high purity electrolyte, in this case NaCl) with the size and number of insoluble particles suspended into the solution (Delmonte et al., 2002).*"
- For Synchrotron radiation (introductory paragraph): "*Additional precautions were adopted to limit contamination and increase the signal to noise ratio. For example, the application of plastic sheets inside the experimental chamber to limit radiation backscattering, the defocus of the incidental beam to illuminate the largest part of the samples or the preservation of high-vacuum conditions during the acquisition, full details are found in Baccolo et al. (2018a).*"
- For X-ray fluorescence synchrotron analysis: "*Major elements in dust were investigated through X-ray fluorescence spectroscopy, using synchrotron radiation as excitation source (Iida, 2013). Samples were irradiated with a 10 keV incident beam (cross section ~1x1 mm) for 600s and the fluorescence signal was acquired with a silicon drift detector, allowing for the quantification of the following elements: Na, Mg, Al, Si, K, Ca, Ti, Mn, Fe. Analytical accuracy was evaluated analyzing NIST standard reference materials (SRM); it decreased from light to heavy elements (standard deviation of the replicates for Na was 25 % of the mean value, 10 % for Fe). Recovery factors were evaluated comparing certified concentrations of SRMs with calculated values: they ranged from 85% to 115% except for Ca and Na (133% and 129%). Full details are given in Baccolo et al. (2018a, b). Elemental concentrations were converted into oxides concentrations and closed to 100%, following an established practice (Rudnick and Gao, 2003).*"
- For X-ray absorption spectroscopy: "*This is a technique relating the spectral features in X-ray absorption spectra to chemical and molecular features of the considered element. To this purpose a sample is irradiated with a beam of monochromatic photons whose energy finely changes with time. The response of the sample depends on elemental features such as oxidation, coordination and mineralogy (Calvin, 2013).*"
- For Relative abundance of Fe-bearing minerals: we think that this part is already sufficiently explained, also considering the supplementary material.

**Rev1:** P. 4 methods: It is described that 54 samples were prepared, the samples per climate period however add up to 55 (l. 93ff). What is the correct number of samples or did I miss something?

**Reply:** The reviewer is right, we made a mistake indicating that we analyzed 3 samples from MIS4, they are actually 2. We have corrected the text.

**Rev1:** A table (possibly in the appendix) providing the formulas of the minerals introduced in 3.3.3. would enhance readability and make it easier to follow the interpretations and weathering effects etc. (4.4.1-4.5). It could even make sense to display the chemical formula of the weathering processes resulting in jarosite, but I leave that to the authors.

**Reply:** we have added the formulas to the section where we discuss the most important minerals (section 4.4 and subsections). We have decided not to add a further table to limit the length of the manuscript. We have also decided not to report a formula describing the reactions leading to jarosite precipitation. The reason is that the process is not simple, as extensively discussed. We have the dissolution of ferrous minerals and then the precipitation of jarosite, but this is something that involves the entire mineral assemblage of dust in TALDICE, not a couple of minerals. We feel that a description is more informative than a sort of simplified scheme that can't be considered a true geochemical formula.

**Rev1:** P. 6 l. 162: I agree that the observed increase in dust size can be due to aggregation. However, "deep ice" is always relative and it would be good to mention that De Angelis et al. (2013) studied much deeper (~2900 m) and older (>400 ky) ice. Furthermore, Simoes et al. (2002) described something similar for Vostok (3450- 3540 m / > 400ky).

**Reply:** We have changed the passage to: "*This can be interpreted as a consequence of dust particle aggregation in deep, a processes that has already been observed in Antarctic ice below 2500 m deep (Lambert et al., 2008; De Angelis et al., 2013).*"

**Rev1:** P. 6 l 168 ff. It´s a very strong statement and other references for other Antarctic ice cores might be needed in addition to Delmonte et al. (2002). Additionally, it is difficult to argue that no post-depositional alteration occurred in EPICA ice since this was not the main objective of Delmonte et al. (2002) (Maybe I missed this explicit statement in the paper though). I suggest to mitigate and rephrase

**Reply:** Modified to: "*Such features are not encountered in surficial sections of Antarctic ice cores, where the effect of post-depositional alterations are limited (Royer et al., 1983; Delmonte et al., 2002; Wegner et al., 2015), they result from in situ aggregation of particles in deep ice (Lambert et al., 2008; De Angelis et al., 2013; Baccolo et al., 2021).*"

**Rev1:** P. 8 l. 181f link Fig. 3 here.

**Reply:** Done

**Rev1:** P. 9 l. 200: Define what is detected for the first time.

**Reply:** Done

**Rev1:** Check wording for "concern", "appreciate", xxxx m deep

**Reply:** Corrected, thanks

**Rev1:** P. 11 l. 221f The cited papers did not analyse Talos Dome ice and it should be stated clearly that only the processes are addressed here. You could add "…reactive species as proposed for Dome Fuji (Ohno et al., 2005) and EDML (Eichler et al., 2019)." Furthermore, the modelled pinning process by Durand et al, 2006 hasn't been observed yet in polar ice (e.g., Faria et al, 2010, Eichler et al, 2017, Stoll et al, 2021). Pinning is likely to play a role in the grain size evolution, but without a dedicated study on the microstructure of MIS 2 from TALDICE we can´t say for sure. Rephrasing to e.g., "probably related to pinning of grain boundaries by insoluble particles as suggested by Durand et al 2006." Could solve this issue. What do you mean with "ice metamorphism is too advanced"? Do you refer to the dynamic recrystallization at 1141 m mentioned in Fig. 2 from Montagnat et al., (2012)?

**Reply:** we agree with the suggestions, now the text reads: "*During these periods the amount of dust deposited at Talos Dome acts as a buffer, neutralizing the acidity of ice and consuming reactive*

*species as proposed for the Dome Fuji and EDML ice cores (Ohno et al., 2005, 2006; Eichler et al., 2019). Figure 4c shows that the growth of ice grains is also temporarily inhibited during MIS2 probably because of grain boundary pinning by insoluble particles as suggested by Durand et al. (2006). Something similar is visible in MIS4, while ice corresponding to MIS6 does not present neither an $Fe^{2+}$ recovery, nor a decrease of ice grain size (Figure 4c), probably because in deep ancient ice in situ oxidation of Fe-minerals and ice re-crystallization are too advanced."*

**Rev1:** I suggest to go through the structure of 4.4. and 4.5 again, the mixture of results and discussion is sometimes difficult to follow. Ways to structure it are e.g., from most abundant (absolute ) to least abundant or relative occurrence with depth. Discussing one mineral per subsection and always starting in the same way could enhance readability, e.g., either first presenting the original origin/process leading to the formation of the mineral or describing your data i.e. how much you observed (I would start with your data first and then discuss the origins and other issues). I would briefly describe Jarosite in 4.4.X (abundance with depth) which can then lead to the "weathering" subsection explaining the origin. I think this link is one of the main strengths of the manuscript and should thus be presented clearly. I support the idea of the weathering index, but would describe it earlier in the text and also briefly in the figure caption (see comment on Fig. 6).

**Reply:** We have decided not to change the general structure of paragraph 4.4 (now paragraph 3.4). We believe that discussing together jarosite and hornblende or siderite and pyrite is better than describing all the minerals one by one. This is because in the discussion we strictly relate the decrease of hornblende to the increase of jarosite and the pair siderite-pyrite behaves very similarly. However we made some changes to the text, trying to make the sub-sections more similar and easy to follow. After the sections dedicated to the minerals, we have now added an additional section (The weathering index) to present the index and describe its behavior. It reads: *"An index was developed to summarize the different pieces of information from the different Fe-minerals. The index is defined the weathering index and it is defined as the % ratio between the rel. ab. of jarosite and the sum between the rel. ab. of jarosite, hornblende, muscovite and hematite. The trend of the index long TALDICE is shown in Figure 6i. It varies between 0 % in the upper part of TALDICE (pristine mineral assemblage) and rapidly increases to 100 % between 1000 and 1300 m, reflecting the progressive weathering of dust, which consist in the consumption of many ferrous minerals and the precipitation of jarosite."* Concurrently we have shortened the section dedicated to the index in paragraph 4.5 (now 3.6) and added some details to the caption of Fig. 6.

**Rev1:** Jarosite is not found in the 3 youngest climate periods displayed in Fig. 5, so please restructure the sentence in P.12 l 233. (e.g., Hornblende and Jarosite are found throughout large parts of the core, Hornblende dominates the young/shallow samples while Jarosite becomes more dominant with depth.)

**Reply:** We have changed to: *"The most evident trends regard hornblende and jarosite (Fig. 6a and g). Hornblende dominates the young/shallow samples while jarosite increases with depth. Since the trends related to these minerals involve large parts of the core regardless of climatic conditions, we interpret them as a consequence of post-depositional processes."*

**Rev1:** 4.4.2 starts with several sentences which could also be part of the introduction of 4.4. It would be easier to follow by keeping the structure, i.e. first stating the results and then discuss them. Are the 3% pyrite and siderite really "relatively common"?

**Reply:** We agree with the comments, we have now modified the paragraph to keep them into account. About the abundance of pyrite and siderite: it must be considered that 3% is a relative abundance which refers to the Fe-mineral fraction of dust. In our samples iron oxides constitute on average 5.5 % of total dust mass, so we have a 3% of 5.5%, surely a common and not anomalous value for both pyrite and siderite. The new version of the paragraph reads: "*Beside the trends involving the entire ice core, some minerals show an additional pattern in correspondence of MIS2. In some cases, it is a relative maximum (muscovite, hematite), in others a minimum (siderite, pyrite). Given the correspondence with MIS2, such features are interpreted as climate-related signals. In the Holocene siderite and pyrite, $FeCO_3$ and $FeS_2$, constitute about 3 % each of Fe-minerals in TALDICE, but during MIS2 they drop to 0.5 %. A partial shift in mineralogy between the Holocene and MIS2 is not unexpected. During MIS2 dust is supplied to Talos Dome mostly by Patagonian sources (Delmonte et al., 2004, 2010), while in the Holocene it is local and comes from Northern Victoria Land (Delmonte et al., 2010; Baccolo et al., 2018b). The presence of siderite and pyrite in Holocene dust at Talos Dome agrees with the geology of Victoria Land, where they are common accessory minerals, owing to the basaltic/doleritic nature of local rocks (Sturm and Carryer, 1970; Dow and Neall, 1974). In addition to the geologic context, also atmospheric transport can partially explain their absence during MIS2. Both minerals are easily oxidized when exposed to the atmosphere. Their lack in MIS2 could be related to their oxidation during the long-range transport from South America. Mineral aerosol is subject to several chemical reactions, and Fe-oxidation is one of the most relevant (Shi et al., 2012). On the contrary, siderite and pyrite survive the transport in the Holocene because of the proximity of local dust sources.*"

**Rev1:** Some statements are made and the used reference are mentioned a sentence or two later (e.g., P. 13 l. 275 "..it has been proposed that additional…"). Naming your references first or connecting the sentences would back up your arguments better.

**Reply:** Changed to: "*During glacial culminations it has been proposed that an additional source other than Patagonia, the Puna-Altiplano dry region in the tropical Andes, supplies dust to Antarctica (Delmonte et al., 2010). In the Puna-Altiplano hematite is widely present (Aubry et al., 1996).*"

**Rev1:** On p. 14 l. 282 it is stated that hematite is not stable in deep ice. However, Hematite was observed e.g., at 2390 m (transition MIS 5.1 to 6) by Eichler et al. (2019) with Raman spectroscopy. To clarify this issue you could simply add "in our analysed TALDICE ice samples" and briefly describe other findings of hematite in deep ice. To explore the processes in deep ice further it would be interesting to know if there is a pH record for TALDICE which could back-up the hypothesis of acidic conditions?

**Reply:** in the paper suggested by the reviewer hematite has actually been identified at 2371 m deep in the EDML ice core but looking at Table 1 of that work it can be seen that only two micro-inclusions showed a Raman-signature compatible with hematite (in total 290 inclusions were analyzed). From that study it seems that hematite can be present in deep ice as trace accessory mineral, it can't be inferred that according to that study hematite is stable in deep ice at other Antarctic sites. For this reason, we prefer not to cite Eichler et al. (2019) here. However, we agree that our findings are related to TALDICE and not in general to deep ice, we have thus reformulated to mitigate the passage. About acidity: unfortunately, an acidity record is not available for TALDICE, however it is important to note that measuring pH of meltwater or of solid ice (the current techniques adopted for ice cores) is completely different than measuring the pH of ice-grain junctions or intra-grain micro-inclusions where weathering takes place (according to our manuscript). I am sure that if we had the possibility to measure the pH of deep TALDICE we would not find strong acidic values. But the identification of jarosite in deep TALDICE is a strong evidence about the presence of local low pH environments in deep TALDICE, otherwise jarosite would have not formed as it requires a low pH to precipitate. The conditions leading to jarosite formation are not compatible with hematite conservation (you can look at the references cited in the manuscript). For this reason we have discussed the absence of hematite in deep ice in this way. New version: "*Hematite, $Fe_2O_3$, is a weathering product in soils under dry and warm conditions, it is typical in tropical regions while it is rarely encountered in cold and wet climates (Schwertmann, 1988). In TALDICE it is mostly found in MIS2, when the dust signature is fully South American (Delmonte et al., 2010). During glacial culminations it has been proposed that an additional source other than Patagonia, the Puna-Altiplano dry region in the tropical Andes, supplies dust to Antarctica (Delmonte et al., 2010). In the Puna-Altiplano hematite is widely present (Aubry et al., 1996). Our results show an excess of hematite during MIS2 (mean rel. ab. 21 %), supporting this hypothesis. A previous study focusing on inner East Antarctica observed a higher abundance of hematite during the Holocene than in MIS2 (Paleari et al., 2019). The difference could be related to the geographic position of the two sites and to the influence of secondary sub-tropical sources during the Holocene. Below 1300 m deep, hematite is not observed, suggesting that this mineral is not stable in deep TALDICE. It is known that under acidic conditions (pH~4) hematite is not stable and can dissolve (Schwertmann and Murad, 1983; Zolotov and Mironenko, 2007), leading to the precipitation of jarosite and goethite (Papike et al., 2006).*"

Moreover, we have added a passage in the paragraph about weathering to better explain the relationship between jarosite precipitation and hematite disappearance: "*The disappearance of hematite in deep TALDICE, which is not stable under acidic conditions (Schwertmann and Murad, 1983), further supports the occurrence of oxidative/acidic weathering in deep ice at Talos Dome.*"

**Rev1**: Think about adding Kuiper et al. (2020b) to the section on pre-melting.

**Reply:** The suggested paper is really interesting (I didn't know it, thanks!) but it doesn't add any relevant information to the manuscript. Since the reference list is already endless, we would like to limit the number of new references and we would prefer not to add it.

**Rev1**: P. 15 l. 320ff. To my knowledge the processes acting on impurities, microstructure and grain growth are not fully understood yet (e.g., Eichler et al. 2019, Stoll et al., 2021). There might be more processes in the ice resulting in the concentration of impurities in addition to large deposition events as suggested for cloudy bands. Which process are you addressing here?

**Reply:** The reviewer is right, the passage was not clear. The new version reads: "*The relationships between ice re-crystallization and the concentration of impurities in deep ice is likely, but it is not yet completely understood (Eichler et al., 2019; Stoll et al., 2021). Our study supports this hypothesis since Fe oxidation and ice grain growth present a very similar trend in TALDICE (Fig. 4c). The weathering evidence recognized in deep TALDICE can only be explained assuming that acidic brines are present in deep ice. Their formation is probably related to the concentration of impurity in localized environments and to the lowering of pressure melting point.*"

**Rev1**: P. 16 l. 328 I suggest to explicitly link Fig. 2 from Baccolo et al. (2021) to the microscopic observations.

**Reply:** we have added the link to the figure

**Rev1**: P. 16 Give some specific examples if possible, e.g., l. 341 "The trend of some elements…", l. 349 "TALDICE presents a number of peculiarities, such as…, if compared to…". Later you can explain them in detail, but a brief overview helps to follow.

**Reply:** We have re-written the paragraph, now it reads: "*TALDICE presents a number of peculiarities if compared to other East Antarctic ice cores: deposition of local dust presenting a basaltic signature (Baccolo et al., 2018a), relatively warm temperatures, and strong influence from the ocean. Such features can partially explain why post-depositional processes affecting dust are so notable at this site. Basaltic-doleritic rocks are easily weatherable, even at low temperature (Li et al., 2016; Niles et al., 2017). In addition, Talos Dome is located near the Southern Ocean and receives a considerable amount of marine aerosol rich in reactive acidic species (Iizuka et al., 2013). A further factor to consider is ice temperature. Talos Dome is near the coast and its climate is tempered by the ocean. When considering the same depth, ice temperature at Talos Dome is about 15-20 °C warmer than at inner sites (Talalay et al., 2020).*"

**Rev1**: I know it is very difficult to define (and to prove) but I would be interested in your thoughts on the exact processes involving high ice-temperature and ice metamorphism altering the dust record. This goes beyond the scope of this study, but is somehow crucial to understand (deep) ice better.

**Reply:** thanks for sharing your thoughts, the review process is also an opportunity to learn and discuss constructively! My feeling is that to deeply understand what happens in deep ice we must identify the conditions allowing for the presence of liquid water (pre-melting). Probably there are some transformations also at shallow temperatures, but they are minor and almost negligible. Once pre-melting starts everything changes and several processes start taking place and involving many different impurities, both soluble and insoluble. Liquid water is the only medium through which chemical reactions and exchanges can become relevant. I am happy that new theoretical studies are coming about pre-melting and ice-recrystallization, but until now they are completely ignoring the role of impurities. We need a lot of experimental data to understand (or at least hypothesize) the mechanisms that drives what I call "englacial geochemistry", with those data it will be hopefully possible to talk with modelist, trying to define a sort of common theory about what happens in deep ice. One thing I am sure of, is that there is still a long way to go!

**Rev1**: Sect. 5 Conclusions and perspectives: The results of this study shine new light on the area of impurities in polar ice. Thus, I suggest so include more details regarding the results/discussion in the

conclusions and to shift the perspectives part to a dedicated subsection of the discussion (e.g., which other elements/minerals would be interesting to investigate deep ice as a geochemical reactor).

**Reply:** we agree that the conclusions didn't properly highlight our main results. We have reformulated the passage, adding some further details. Here the new version: "*This study provides a first description of dust chemical weathering in deep polar ice. Grain size, concentration, mineralogy and composition of dust are all affected by post-depositional processes. Fe speciation and mineralogy are efficient probes to explore such transformations in Antarctic ice. The englacial precipitation of jarosite, the oxidation of Fe in dust, the decline of ferrous minerals (hornblende, pyrite, siderite, muscovite) and the depletion of some major elements (Ca, Mg, Na) suggest that below 1000 m deep, dust in TALDICE is affected by acidic-oxidative weathering resulting from the interaction with highly saline and acidic brines. The production of such brines is likely related to ice re-crystallization and to the accumulation of impurities in highly localized environments where reactions between soluble and insoluble species are favored. From this perspective deep Antarctic ice can be seen as a "geochemical reactor", capable of promoting the precipitation of secondary minerals and the concurrent dissolution of others*."

However, we would like to keep the perspectives into the final paragraph. This is the first work (to the best of our knowledge) which presents deep Antarctic ice as an environment where peculiar geochemical reactions take place. The aim of this work is not only to share our results but also to encourage future studies on this topic which is still poorly considered within the ice core community. For this reason we believe that keeping in the final paragraph some recommendations for future studies could help to spread englacial geochemistry.

We have kept into account all the technical/specific comments from the reviewer, with the exceptions of these ones:

**Rev1**: P. 3 l. 56 "secondary Fe-bearing minerals", maybe give 1 or 2 examples
**Reply:** in the mentioned paper the authors couldn't recognize the nature of the Fe-concretions observed on the surface of insoluble inclusions, they interpreted them in the light of their micro-morphology and because of their Fe-rich composition, not compatible with primary mineral aerosol. For this reason we can't report detailed examples here.

Thank you very much for your thorough review, it really helped to improve the quality of the manuscript.

Best regards,

Giovanni Baccolo and coauthors

---

## Author Comment (AC2)

Replies to reviewers:

Reviewer #2

**Rev2:** The authors present a new perspective of possible post-depositional processes that affect mineral dust records, particularly below 1000 m depth in the 1620 m long TALDICE ice core. The datasets are of high quality and obtained using Coulter counter and spectroscopic measurements such as Synchrotron radiation, X-ray fluorescence and XANES. Crucial properties of dust such as the concentration, grain-size, their elemental composition with a focus on Fe-mineralogy are discussed in depth. The study shows englacial formation of specific minerals in deep ice affecting the original scenario of dust deposition and conclude highlighting potential impacts while interpreting dust records on deep ice cores. While the originality, scientific quality and significance of the work is excellent, the manuscript falls short on language with several grammatical errors which needs language editing. I recommend this manuscript, after necessary language editing, for publication in the journal's special issue: Oldest ice: finding and interpreting climate proxies in ice older than 700000 years.

**Reply:** thank you very much for this positive comment and for your suggestions which will improve the quality of our manuscript

**Rev2:** I do not mention any corrections regarding language as the manuscript needs thorough language editing. Following are some specific comments.

**Reply:** in the new version of the manuscript an accurate revision of language and grammar will be carried out.

**Rev2:** Line 35: The authors mention the role of depth and pressure in the post-depositional processes that has not been previously addressed - however this aspect of depth and pressure altering dust records aren't discussed in the results and discussion. I suggest to modify or delete this sentence.

**Reply:** agreed

**Rev2:** Line 38: replace ine with ice.

**Reply:** corrected

**Rev2:** Section 2: Though co-ordinates are provided, I suggest a location map especially with surrounding dust sources would be useful for many.

**Reply:** we have updated Fig.1 adding a small and simple map to show the position of Talos Dome in Antarctica. We didn't add information about local dust sources, this topic is extensively discussed in some of the references cited in the text (Delmonte et al., 2010; Baccolo et al., 2018).

**Rev2:** Sample Preparation: I have some queries on technical aspects of sample preparation. You mention that the preparation took place in the ISO6 clean room - were the ice sections decontaminated under the laminar flow bench or in the clean room? At what temperature did this process take place? Also, considering the 2 cm thin ice sections used in this analysis, how thin was the ice after 3 baths decontamination? do you decontaminate the ice sections using ice cold ultra-pure water bath to avoid melting the sections that are already 2 cm thin?

**Reply:** the preparation took place inside the clean room (ISO6), in particular the decontamination was carried out on a table available in the room, but not directly under the bench. Once decontaminated, samples were placed under the laminal flow of the ISO5 bench. The procedure took place at ambient air temperature, which inside the room is set constant at 18°. About the ultra-pure water baths: each bath takes about 20 seconds. From our experience this is a good compromise which allows to have a sufficiently large sample with a sufficient removal of the outer ice layers. At the end of the three baths the ice strips looses about half of their mass and thickness goes from 2 cm to a little bit more than 1 cm. During the procedure ice is always handled with pre-cleaned plastic forceps.

**Rev2:** I understand that you analysed 191 coulter counter samples and 54 filtered samples for spectroscopy. If not, you might have to clarify it in the sample preparation section.

**Reply:** yes, the reviewer is right

**Rev2:** line 78: remove extra "in".

**Reply:** done

**Rev2:** Line 93: According to this paragraph, there are 55 samples, while you mention 54 in Lines 78 and 87. Your dataset in the supplement seems to have 54 samples.

**Reply:** thanks for highlighting this mistake. The samples are actually 54, we made a mistake saying that we prepared 3 sample from MIS4, they are actually 2.

**Rev2:** Line 144: CLPP (coarse local particle percentage).

**Reply:** corrected.

**Rev2:** Lines 197–201: This paper focuses on many possible reactions of Fe-minerals happening in deep ice. The authors do mention about carbonate dissolution in deepest samples backed with well-known ice core studies. However, there is also a possibility that such post-deposition processes alter dust chemistry immediately after the snow deposition as shown from the surface snow cores by Mahalinganathan and Thamban (2016) that has not been observed in the holocene / interglacials of deep ice cores. Do you think the carbon dissolution and Fe-mineral reactions which are apparent in deep sections of TALDICE may be happening constantly from the time after snow deposition (instead of happening at a deeper section, (unless it is depth-pressure based), but are missed due to lesser spatial study?

**Reply:** thanks, I didn't know that paper, it was an interesting reading. We are quite sure that dust deposited in Antarctica, especially if transported from extra-Antarctic sources, has already undergone to reactions and changes during the atmospheric transport. A partial dissolution of carbonates during transport is likely, in particular during glacials. However, the processes that we are describing in the Talos Dome ice core has very clear trends which involve the ice below a certain threshold. From our evidence it is evident that the precipitation of jarosite is not favored in the first 1000 m of the core. At the same time, we note that Fe-oxidation (Fig. 1c) starts at about 500 m deep, as also the depletion of Ca and Mg in mineral particles (see

Fig. S4). This means that Fe reaction and the precipitation of secondary minerals are possible only below a threshold. It remains to be understood if the threshold is related to pressure, temperature, time, ice re-crystallization or probably to a combination of these processes.

**Rev2:** Figure 1: ngdustg−1 ice of dust concentration in figure. The caption misses mentioning MIS 7.5 and 9.5 for red bands.

**Reply:** corrected.

**Rev2:** Figure 3: Reference is not linked.

**Reply:** we have now added the missing reference

**Rev2:** Figure 6: Choose contrasting colors for panels c, d and e.

**Reply:** we have updated the figure accordingly

**Rev2:** Table 2: SD for MIS-6 column is missing

**Reply:** this is because only one sample corresponds to this climatic stage and we couldn't determine SD

Thank you very much for your careful reading and for your constructive suggestions

Best regards,

Giovanni Baccolo and the coauthors